# MODEL UNMERGING: MAKING YOUR MODELS UNMERGEABLE FOR SECURE MODEL SHARING

## ABSTRACT

Model merging leverages multiple finetuned expert models to construct a multi-task model with low cost, and is gaining increasing attention. However, as a growing number of finetuned models become publicly available, concerns about the safety of model merging have emerged. Unauthorized merging may infringe on developers' rights and risk leaking sensitive personal information. Most existing methods focus on detecting whether a merged model originates from a specific source model, but fail to effectively prevent illegal merging. In this paper, we propose *MergeLock*, an active protection mechanism that disrupts model parameters to render them unmergeable, thereby directly preventing unauthorized model merging. Specifically, leveraging the inherent symmetry of the attention mechanism in Transformer-based models, we randomly sample two pairs of invertible matrices and apply them to the Query-Key (QK) and Value-Output (VO) branches. This transformation keeps the model's output unchanged while pushing it away from the shared parameter space of other finetuned models. Extensive experiments across both vision and language tasks demonstrate that *MergeLock* can degrade the performance of merged models by over 95% when a protected model is involved in most cases, demonstrating its effectiveness. Moreover, we further demonstrate that merged models protected by *MergeLock* cannot be effectively recovered using low-cost restoration methods, further enhancing robustness against unauthorized merging.

## 1 INTRODUCTION

Multi-task learning (MTL) enables a single model to handle multiple tasks simultaneously, in contrast to training separate models for each task. This shared architecture significantly reduces storage requirements and improves inference efficiency (Caruana, 1997; Vandenhende et al., 2022; Zheng et al., 2023). Owing to these advantages, MTL has gained popularity in various domains, including computer vision (Chen et al., 2018; Yang et al., 2023; Liu et al., 2019), natural language processing (Collobert & Weston, 2008; Dong et al., 2015), recommendation systems (Ma et al., 2018; Tang et al., 2020), and speech recognition (Ravanelli et al., 2020; Zhao et al., 2019). However, traditional MTL methods require training on all relevant datasets, which complicates data collection and poses challenges for stable training, particularly when handling a large number of tasks simultaneously.

Model merging (Wortsman et al., 2022; Matena & Raffel, 2022; Tang et al., 2024; Yang et al., 2024a) has emerged as a promising alternative to traditional MTL methods, aiming to address their limitations. Instead of training a unified model from scratch on all tasks, model merging constructs a multi-task model by directly manipulating the parameters of independently trained, task-specific models. This approach eliminates the need for extensive data collection and retraining, thereby greatly reducing computational and storage overhead. The most straightforward merging method is weight averaging (Utans, 1996), which simply averages the parameters of multiple models to obtain a merged model. Another representative technique is task arithmetic (Ilharco et al., 2023), which constructs a task vector by computing the difference between a fine-tuned model and its corresponding pretrained model, and then applies these vectors to create new models for downstream tasks. Recent state-of-the-art methods, such as Ties-Merging (Yadav et al., 2023a), AdaMerging (Yang et al., 2024c), Dare (Yu et al., 2024), WEMoE (Shen et al., 2024) and TSV-M (Gargiulo et al., 2025), are all built upon the task arithmetic framework, offering improved compatibility and perfor-

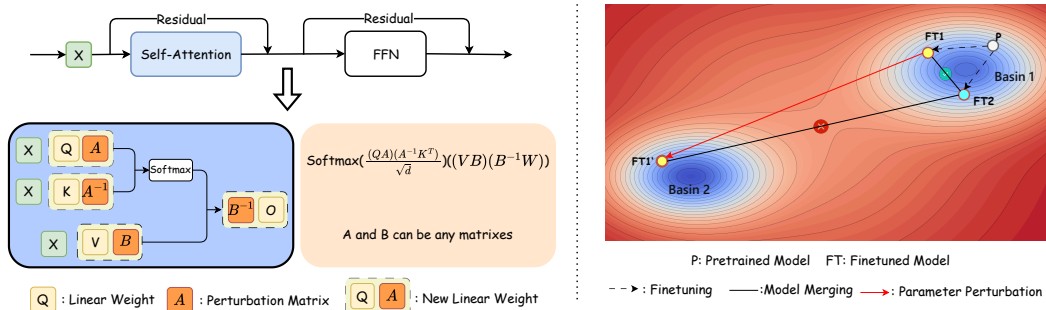

Figure 1: **(a) Left**: To prevent model merging, transformation matrices $A, A^{-1}, B, B^{-1}$ are inserted into the query (Q), key (K), value (V) and output (O) branches of the self-attention layers. This transformation disrupts the parameter space to block effective merging, while preserving the original model's output equivalence. **(b) Right**: Visualization of our unmergeable strategy in the loss landscape. Normally, two finetuned models (FT1, FT2) derived from the same pretrained model (P) lie within the same loss basin 1, and their merged model (✓) achieves low loss. However, after applying our unmergeable transformation to FT1, the model (FT1′) is pushed into a different basin 2. As a result, merging FT1′ with FT2 produces a model (×) that falls into a high-loss region.

mance in complex multi-task settings. As more models become publicly available, model merging is expected to play an increasingly critical role across a wide range of domains and applications.

As model merging continues to gain traction, several critical challenges have emerged. A primary concern is that open-source models are becoming increasingly vulnerable: since model merging is both easy to perform and difficult to detect or trace, it poses a significant threat to the intellectual property (IP) of model developers (Cong et al., 2024). Moreover, model merging can compromise safety alignment; maliciously trained models may exploit the merging process to extract personally identifiable information (PII) or infer membership information (MI), thereby introducing serious privacy risks (Guo et al., 2025). Although existing techniques such as watermarking (Li et al., 2023a) and fingerprinting (Cao et al., 2021) have been proposed, they primarily serve as detection mechanisms rather than preventive solutions (Cong et al., 2024), leaving unauthorized model merging largely unprevented. Therefore, there is an urgent need for preventive solutions against unauthorized model merging.

In this paper, we propose *MergeLock*, a method designed to make models unmergeable, i.e., resistant to model merging, without compromising their original performance. Under this protection scheme, any unauthorized attempt to merge the protected model with others will result in a severely degraded model that performs poorly across all evaluation datasets. Specifically, inspired by the existence of multiple equivalent parameter spaces (basins) in deep neural networks (Entezari et al., 2022; Ainsworth et al., 2023), we apply an equivalent transformation to the parameters of the self-attention layers (see Fig. 1(a)). These layers are a core component of Transformer-based architectures. This transformation moves the protected model away from the shared loss basin typically occupied by other fine-tuned models originating from the same pretrained model (see (FT1′ & FT2) vs. (FT1 & FT2) in Fig. 1(b)). When an unmergeable model is merged with a regular model, the merged model exhibits a sharp increase in loss due to the significant discrepancy between their parameter spaces.

Extensive experiments across both vision and language tasks demonstrate that *MergeLock* effectively degrades the performance of merged models. Even when parameter alignment techniques are applied to the protected model's parameters, it remains difficult to recover acceptable performance. More specifically, in our experiments, the model protected by *MergeLock* degrades the overall performance of the merged model by approximately 95% , and the application of an alignment method recovers only about 5% of the original performance. One additional advantage of our method is that, developers retain full control over the distribution of protected models: by sharing a secret key (i.e., transformation matrices), authorized users can restore the original model.

To summarize, the main contributions of this paper are in three aspects:

- **Theoretical Analysis:** We analyze the symmetry properties of Feedforward Networks (FFNs) and self-attention layers in Transformer models, and demonstrate that self-attention layers are more effective than FFNs in terms of parameter perturbation.

- **Novel Protection Method:** Leveraging the inherent symmetry of self-attention, we propose *MergeLock*, a novel method that renders models unmergeable, thereby protecting intellectual property and preventing unauthorized model merging.
- **Comprehensive Validation:** Extensive experiments demonstrate the effectiveness of *MergeLock*: merging an unmergeable model with any other model leads to the merged model with severely degraded or non-functional performance. Even with advanced alignment techniques, recovery is infeasible without data or significant computation.

## 2 RELATED WORK

**Model Merging.** Model merging combines multiple models with the same architecture to produce a more powerful model (Li et al., 2023b). It offers high flexibility: even when models are trained on the same dataset, those with different training configurations or at various training stages can be merged to enhance utility or generalization (Izmailov et al., 2018; Gupta et al., 2020; Cha et al., 2021). For example, ModelSoup (Wortsman et al., 2022) greedily selects and averages models trained under varying settings, yielding improved performance. When models are trained on different datasets or tasks, merging can also produce a unified model capable of handling multiple tasks. Data-free methods such as Task Arithmetic (Ilharco et al., 2023), Ties-Merging (Yadav et al., 2023b), DARE (Yu et al., 2024), Consensus TA (Wang et al., 2024), ISO (Marczak et al., 2025) and TSV-M (Gargiulo et al., 2025) exploit structural similarities in parameters to achieve effective merging. Data-driven methods like Fisher Merging (Matena & Raffel, 2022), RegMean (Jin et al., 2023), AdaMerging (Yang et al., 2024c), Surgery (Yang et al., 2024b), and AdaRank (Lee et al., 2025) further improve performance through guidance from training data or unlabeled testing data. However, as model merging techniques gain popularity, concerns about model security and unauthorized merging have also emerged.

**Defense Against Unauthorized Model Merging**. Model merging introduces several security and ethical risks, including unauthorized model reuse (Cong et al., 2024), safety misalignment, and unintended information leakage (Guo et al., 2025). To address these concerns, recent works have proposed two main categories of defense mechanisms: model detection and model protection. *Model detection* methods, such as watermarking (Adi et al., 2018) and fingerprinting (Xu et al., 2024), embed identifying information into the model via fine-tuning. Even after merging, the model can still output predefined responses or preserve unique behaviors, which helps detect unauthorized use. In contrast, *model protection* aims to prevent merging from succeeding in the first place. These methods directly manipulate model parameters to make merging ineffective, while preserving the model's original performance. Compared to detection, protection provides a more proactive defense by fundamentally breaking the compatibility assumptions required for merging. Our work focuses on model protection. The most related method is PaRaMS (Wei et al., 2025), which applies permutation matrices to MLP layers. It uses the Hungarian algorithm to maximize the discrepancy between the protected and original models. While PaRaMS can significantly hinder merging, it is vulnerable to attacks: in many cases, merging recovery can restore up to 95% of the performance. In contrast, our *MergeLock* targets self-attention layers, leveraging their structural symmetry to achieve strong unmergeability. Moreover, due to the nature of the transformations, it is considerably harder to reverse or align using purely mathematical methods.

## 3 METHODOLOGY

We begin by introducing the notation and formally defining the model unmerging problem in Sec. 3.1, along with the notion of symmetry in neural network parameter spaces in Sec. 3.2. Next, we analyze the differences in symmetry properties between feedforward networks and self-attention layers in Sec. 3.3, highlighting the advantages of targeting self-attention for unmergeable model construction. Lastly, in Sec. 3.4, we present our proposed method, *MergeLock*, which leverages the symmetry in self-attention layers to create unmergeable models that maintain their original performance while resisting unauthorized merging attempts.

## 3.1 PRELIMINARIES

**Model Merging.** Model merging aims to combine several independently fine-tuned models into a unified one to enhance generalization across multiple tasks (Li et al., 2023b). This process can be formalized as $\theta_m = \mathcal{M}(\theta_{\text{pre}}, \theta_1, \ldots, \theta_n)$, where each $\theta_i$ is derived from a common pre-trained model $\theta_{\text{pre}}$ via fine-tuning on a specific dataset $\mathcal{D}_i$. The merged model $\theta_m$ is expected to inherit all task capabilities from the individual models, i.e.,

$$\mathbb{E}_{(x,y)\sim\mathcal{D}_i}[\mathcal{L}_i(f(\theta_m, x), y)] \approx \mathbb{E}_{(x,y)\sim\mathcal{D}_i}[\mathcal{L}_i(f(\theta_i, x), y)], \text{ s.t. } \forall i \in [n], \tag{1}$$

where $f(\theta, x)$ denotes a neural network parameterized by $\theta$, and $\mathcal{L}_i()$ is a task-specific loss function. $\mathcal{M}()$ denotes the merge algorithm. **Task Arithmetic (TA)** (Ilharco et al., 2023) is a classical model merging approach, upon which many advanced methods build. TA defines the task vector as $\tau_i = \theta_i - \theta_{\text{pre}}$, and constructs the merged model as: $\theta_m = \theta_{\text{pre}} + \lambda \sum_{i=1}^{n} \tau_i$, where $\lambda$ is a scaling coefficient that controls the merge strength. The merged model $\theta_m$ is expected to perform well on all tasks $\mathcal{D}_i$.

**Model Unmerging.** In scenarios where a model $\theta_i$ is trained on proprietary or sensitive data, it is often desirable to prevent unauthorized model merging. Motivated by the presence of multiple equivalent basins in the loss landscape of deep neural networks, our objective is to modify the model parameters in such a way that the original performance is preserved, while making the model resistant to common merging strategies. We denote the transformed model as $g(\theta_i)$, which is considered unmergeable if it satisfies the following two conditions:

- *Performance Preservation Condition:* The transformed model $g(\theta_i)$ should maintain the original performance on the task $\mathcal{D}_i$:

$$\mathbb{E}_{(x,y)\sim\mathcal{D}_i}[\mathcal{L}_i(f(g(\theta_i), x), y)] = \mathbb{E}_{(x,y)\sim\mathcal{D}_i}[\mathcal{L}_i(f(\theta_i, x), y)], \tag{2}$$

- *Unmergeability Condition:* The transformed model $g(\theta_i)$ should not be easily merged with other models $\theta_j$ (where $\forall i, j \in [n], j \neq i$) to recover the original task performance:

$$\mathbb{E}_{(x,y)\sim\mathcal{D}_i}[\mathcal{L}_i(f(\mathcal{M}(\theta_{\text{pre}}, g(\theta_i), \theta_j), x), y)] \gg \mathbb{E}_{(x,y)\sim\mathcal{D}_i}[\mathcal{L}_i(f(\mathcal{M}(\theta_{\text{pre}}, \theta_i, \theta_j), x), y)], \tag{3}$$

Equations 2 and 3 imply that merging any $\theta_j(j \neq i)$ with the protected model $g(\theta_i)$ significantly degrades performance, even though $g(\theta_i)$ itself remains functionally identical to its original version. This is crucial for protecting proprietary models from unauthorized merging attempts.

## 3.2 SYMMETRY IN NEURAL NETWORK PARAMETER SPACES

The Transformer architecture (Vaswani et al., 2017) has become the dominant backbone in modern deep learning, and our work focuses specifically on models built upon this framework. Recent studies have revealed various symmetry properties in Transformer-based models, including those in FNNs and self-attention layers (Godfrey et al., 2022; Navon et al., 2023; Zhao et al., 2025). Understanding these symmetries is essential for designing robust and unmergeable models. In the following sections, we first review the symmetry properties of FNN and self-attention components. Building on these insights, we then introduce our method, *MergeLock*, which leverages reversible transformations to preserve functional equivalence while enhancing model unmergeability.

**Symmetry in Feedforward Networks.** In most Transformer-based models, the FFN is implemented as a two-layer multilayer perceptron (MLP), formulated as:

$$\text{MLP}(X) = \sigma(XW_1^\top + b_1)W_2^\top + b_2, \tag{4}$$

where $\sigma(\cdot)$ is an element-wise activation function (e.g., ReLU (Agarap, 2018), Sigmoid (Rumelhart et al., 1986), Tanh (LeCun et al., 1998)). Here, $X \in \mathbb{R}^{T \times d_{\text{h}}}$ denotes the input sequence representation with sequence length $T$ and hidden size $d_{\text{h}}$. $W_1 \in \mathbb{R}^{d_{\text{f}} \times d_{\text{h}}}$ and $b_1 \in \mathbb{R}^{d_{\text{f}}}$ are the weight matrix and bias of the first linear projection, mapping the input to an intermediate feedforward dimension $d_{\text{f}}$ (typically $4 \times d_{\text{h}}$). $W_2 \in \mathbb{R}^{d_{\text{h}} \times d_{\text{f}}}$ and $b_2 \in \mathbb{R}^{d_{\text{h}}}$ are the parameters of the second projection, mapping back to the model dimension.

Due to the element-wise nature of the activation, applying a permutation matrix before the activation does not alter the output values, but merely permutes their positions. This introduces symmetric structures in FFNs that can be exploited for transformation without affecting functional outputs. Based on this property, symmetric transformations can be applied as follows:

$$W_1' = P^\top W_1, b_1' = b_1 P, W_2' = W_2 P, \tag{5}$$

The following derivation shows that, after applying the permutation, the output of the MLP remains unchanged:

$$\text{MLP}'(X) = \sigma(X{W_1'}^\top + b_1')W_2'^\top + b_2 = \sigma(XW_1^\top P + b_1 P)P^\top W_2^\top + b_2$$
$$= \sigma(XW_1^\top + b_1)PP^\top W_2^\top + b_2 = \sigma(XW_1^\top + b_1)W_2^\top + b_2 = \text{MLP}(X),$$

(6)

where $P \in \mathbb{R}^{d_f \times d_f}$ is a permutation matrix, which is a square matrix with exactly one entry of 1 in each row and each column and 0s elsewhere. This property forms the basis of (Wei et al., 2025), which uses permutations and the Hungarian algorithm to maximize parameter distance across models. However, such protection can be *reversed by simply applying the inverse permutation*, which is trivial to compute.

**Symmetry in Self-Attention Layers.** In feedforward layers, the use of non-linear activation functions necessitates careful handling of matrix transformations, specifically, the permutation matrix $P$ must commute with the element-wise activation $\sigma(\cdot)$. In contrast, self-attention layers do not introduce such element-wise non-linearities between linear projections and attention operations, which allows for more flexible transformations.

Self-Attention mechanism is a core component of Transformer architectures, enabling the model to focus on different parts of the input sequence. A standard self-attention layer can be formulated as:

$$\text{ATTN}(X) = \text{Cat}_{h=1}^H \{X_{\text{QKV}}^h\}W_O^\top + b_O, \text{ where } X_{\text{QKV}}^h = \text{Softmax}(X_Q^h(X_K^h)^\top/\sqrt{d_k})X_V^h,$$

(7)

where $\text{Cat}\{\cdot\}$ denotes concatenation across $H$ attention heads, and $X_Q^h$, $X_K^h$, $X_V^h$ are the query, key, and value representations of the $h$-th head, respectively. $b_O$ is the output bias. Here, $W_Q, W_K, W_V \in \mathbb{R}^{d_k \times d_h}$ denote the projection matrices mapping the input $X \in \mathbb{R}^{T \times d_h}$ to query, key, and value spaces, respectively, and are typically partitioned into $H$ head-specific projections $\{W_Q^h, W_K^h, W_V^h\}_{h=1}^H$, each of shape $\mathbb{R}^{d_k \times d_{\text{head}}}$, where $d_{\text{head}} = d_h/H$. Similarly, $W_O \in \mathbb{R}^{d_k \times d_h}$ is the output projection, partitioned into $\{W_O^h\}_{h=1}^H$ with $W_O^h \in \mathbb{R}^{d_{\text{head}} \times d_h}$.

As shown in Fig. 1, self-attention layers exhibit symmetry properties. For each attention head, we can insert a pair of invertible matrices (i.e., $A$ and $A^{-1}$) to transform the parameters while keeping the output unchanged. Specifically, we can rewrite the query and key projections as:

$$X_Q^h(X_K^h)^\top = (X(W_Q^h)^\top + b_Q^h)(X(W_K^h)^\top + b_K^h)^\top = (X(W_Q^h)^\top + b_Q^h)AA^{-1}(X(W_K^h)^\top + b_K^h)^\top$$
$$= (X(A^\top W_Q^h)^\top + b_Q^h A)(X(A^{-1}W_K^h)^\top + b_K^h A^{-1})^\top,$$

(8)

where $A$ is any invertible matrix of appropriate dimensions. Similarly, we can transform the value and output projections. Let $W_O$ be partitioned into $H$ output heads $W_O^h$. Then, each head satisfies:

$$X_V^h(W_O^h)^\top = (X(W_V^h)^\top + b_V^h)(W_O^h)^\top = (X(W_V^h)^\top + b_V^h)BB^{-1}(W_O^h)^\top$$
$$= (X(B^\top W_V^h)^\top + b_V^h B)(W_O^h B^{-1})^\top,$$

(9)

where $B$ is another invertible matrix of appropriate dimensions. By applying these transformations, we can construct a new self-attention layer with transformed parameters: $\text{ATTN}'(X)$, which is computed using the transformed query and key projections from Equation 8 and the transformed value and output projections from Equation 9. The output remains unchanged, i.e., $\text{ATTN}'(X) = \text{ATTN}(X)$. This property is crucial for maintaining the functional equivalence of the model while introducing transformations that enhance unmergeability.

## 3.3 RETHINKING SYMMETRY IN MLPs AND SELF-ATTENTION LAYERS

The symmetry properties of FFN and self-attention layers differ significantly, which has important implications for model unmergeability. Compared to the symmetry properties in FFN, self-attention layers offer several advantages for transformation-based protection strategies:

- **Structural Consistency.** Self-attention layers exhibit a highly consistent architecture across Transformer variants, typically consisting of multi-head attention followed by a unified output projection. In contrast, MLP/FFN structures vary more widely; for example, some employ Gated Linear Units (GLU)(Shazeer, 2020) or Mixture-of-Experts (MoE) (Lepikhin et al., 2020), which introduce element-wise gating or dynamic routing, making it harder to design general symmetry-preserving transformations.

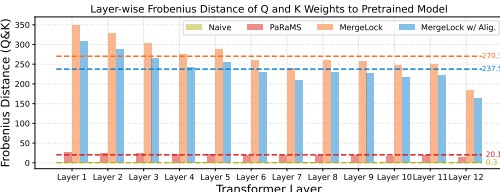 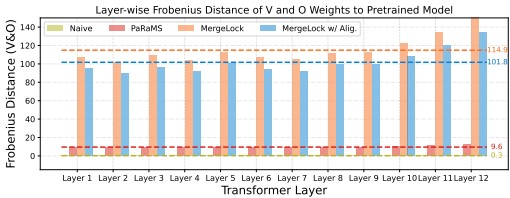

Figure 2: (**left**) Frobenius distance of Q&K weights to pretrained model on SUN397. (**right**) Frobenius distance of V&O weights to pretrained model on SUN397.

- **Activation Constraints.** In MLP layers, the presence of non-linear activations (e.g., ReLU, GELU, or custom variants) constrains valid transformations to *discrete* ones such as permutations, since arbitrary continuous transformations may alter the activation outputs. By contrast, self-attention layers are purely linear before the softmax operation, enabling the use of a broader class of *continuous* transformations (e.g., rotations, scaling, or general invertible matrices) without breaking functional equivalence.
- **Expressive Transformations.** The ability to apply general invertible matrices in self-attention layers significantly increases the expressive space of transformations. This flexibility allows the combination of *discrete* disturbances (e.g., head permutations) with *continuous* disturbances (e.g., orthogonal rotations or non-orthogonal invertible mappings), resulting in stronger and more robust unmergeable model constructions compared to MLP-based permutations.

These advantages make *self-attention layers a more suitable target for constructing unmergeable models*. By leveraging the inherent symmetry properties of self-attention, we can design transformations that effectively disrupt the alignment of finetuned models while preserving their original performance, as detailed in the next section.

### 3.4 MODEL UNMERGING: *MergeLock*

The effectiveness of model merging largely stems from the observation that, under the pre-train–finetune paradigm, models fine-tuned from the same pre-trained checkpoint often converge to a shared or closely aligned basin in the loss landscape (Zhou et al., 2024). As a result, these fine-tuned models are typically very close in terms of Frobenius distance between parameters (e.g., FT1 and FT2 in Fig. 1), which grants them desirable properties such as *linear mode connectivity* (Entezari et al., 2022; Ainsworth et al., 2023).

Our *MergeLock* aims to break this alignment by relocating the model to a distinct basin in parameter space. Concretely, we apply the transformations described in Equations 8 and 9, using different transformation matrices for different layers and attention heads. Each transformation matrix $A$ in Equation 8 (or $B$ in Equation 9) is constructed as the product of three components: $A = RPD$, where each component in the transformation serves a distinct purpose:

- Component $R$ is a random matrix that introduces stochastic perturbations, increasing the diversity and unpredictability of the transformed parameter space.
- Component $P$ is a permutation matrix that reorders the parameter dimensions, disrupting structural alignment between models while preserving functional equivalence.
- Component $D$ is a diagonal scaling matrix that independently scales each dimension, further enlarging the distance between models in parameter space without affecting the model's output.

The combination of these three matrices ensures that the transformed model remains functionally identical to the original, yet is relocated to a distinct and unpredictable region in parameter space, thereby preventing effective model merging.

**Discussion**. Compared to the PaRaMS method in Wei et al. (2025), which applies only the diagonal scaling matrix $D$ to self-attention layers, this strategy alone is insufficient to introduce a substantial discrepancy between models. As shown in Fig. 2, the Frobenius distance between the transformed parameters of two models remains relatively small in PaRaMS (i.e., 20.1 for $Q\&K$ and 9.6 for $V\&O$). In contrast, our *MergeLock* significantly increases the Frobenius distance between the transformed parameters of two models (i.e., 270.3 for $Q\&K$ and 114.9 for $V\&O$), making them unmergeable. We will discuss this part in detail in the experimental section.

## 4 EXPERIMENTS

In this section, we present experiments to evaluate the effectiveness of *MergeLock* in preventing model merging. Sections 4.1 and 4.2 in the main text describe the experimental setup and evaluate the protection effect of *MergeLock* by merging the protected model with another fine-tuned model. Additional experiments, including the investigation of alignment strategies and the analysis of parameter alterations after transformation, are provided in the appendix.

### 4.1 EXPERIMENTAL SETUP

**Models.** In our main experiments, we primarily use CLIP-ViT models (Radford et al., 2021) for image classification tasks. These models are widely adopted in previous studies on model merging (Ilharco et al., 2023; Yadav et al., 2023b; Yang et al., 2024c), making them strong and representative base models for evaluating the effectiveness of our method. We also conduct experiments on Flan-T5 (Chung et al., 2024) for text-to-text generation tasks in the Appendix E to demonstrate the generality of our method across different architectures.

**Datasets.** We conduct experiments on eight widely-used image classification datasets: SUN397 (Xiao et al., 2016), Cars (Krause et al., 2013), RESISC45 (Cheng et al., 2017), EuroSAT (Helber et al., 2019), SVHN (Netzer et al., 2011), GTSRB (Stallkamp et al., 2011), MNIST (Deng, 2012), and DTD (Cimpoi et al., 2014). These datasets cover a diverse range of image classification tasks and allow us to comprehensively evaluate the performance of our method across various domains.

**Merging Methods.** We adopt Task Arithmetic (Ilharco et al., 2023) as the primary merging strategy, as it serves as the foundation for many advanced merging approaches. We also evaluate our method using Ties-Merging (Yadav et al., 2023b) and AdaMerging (Yang et al., 2024c) in the Appendix E.

**Protecting Methods.** In addition to our proposed method *MergeLock*, we use PaRaMS (Wei et al., 2025) as a strong baseline. PaRaMS applies the transformation described in Equation 6 to the FNN and introduces two pairs of randomly sampled diagonal matrices into the attention layers. For the FNN, it employs the Hungarian algorithm to maximize the parameter distance between the fine-tuned and pretrained models. For further details on PaRaMS, please refer to Appendix B.

### 4.2 MAIN RESULTS

This section evaluates the protection effect of *MergeLock* from two perspectives: (1) the effectiveness of preventing model merging in Sec. 4.2.1, (2) robustness against alignment-based recovery attempts in Sec. 4.2.2.

#### 4.2.1 EVALUATION OF PROTECTION EFFECTIVENESS

In this subsection, we assess the effectiveness of *MergeLock* in preventing model merging. We follow the experimental protocol established in PaRaMS (Wei et al., 2025). More specifically, we consider merging two fine-tuned models using Task Arithmetic. One of the models (vertical) is protected using either *MergeLock* or PaRaMS, while the other model (horizontal) remains unprotected. We then evaluate the performance of the merged model on the two respective tasks. In addition, we also evaluate the scenario where two normal fine-tuned models (i.e., without any protection) are merged as a baseline (upper bound).

**Performance Comparison.** As shown in Table 1, we report the classification accuracy of the merged models on eight datasets, where each cell indicates the accuracy of merging a model fine-tuned on the dataset in the row with another model fine-tuned on the dataset in the column. The diagonal cells are marked as "NA" since merging two models fine-tuned on the same dataset is not applicable. The last two columns present the average accuracy of the merged models in each row and the performance drop compared to the baseline (i.e., merging two normal models). We can draw the following conclusion: (1) For "Normal" baseline, the accuracy of merging two clean models (the first row of each block) is generally high, indicating that Task Arithmetic is effective in integrating knowledge from different tasks. (2) Compared to the baseline, merging a *MergeLock*-protected model with a normal model (the fourth row of each block) results in a significant drop in accuracy,

Table 1: Classification accuracy (%) of ViT-B/32 models merged via task arithmetic (TA). Avg. denotes the average of the merged models in each row; in MergeLock, $\Delta$ is the performance gap w/ vs. w/o the protection mechanism; in MergeLock w/ Alig., $\Delta$ is the gap w/ vs. w/o alignment. PaRaMS and PaRaMS w/ Alig. follow similarly.

| | | SUN397 | Cars | RESISC45 | EuroSAT | SVHN | GTSRB | MNIST | DTD | Avg. | $\Delta$ |
|---|---|---|---|---|---|---|---|---|---|---|---|
| SUN397 | Normal | | 64.7 | 80.8 | 84.4 | 83.0 | 81.7 | 84.8 | 67.9 | 78.1 | |
| | PaRaMS | | 1.4 | 6.4 | 10.4 | 10.7 | 3.9 | 6.6 | 2.4 | 5.9 | ↓72.2 |
| | PaRaMS w/ Alig. | NA | 64.0 | 80.7 | 84.3 | 82.7 | 81.4 | 84.7 | 67.0 | 77.8 | ↑71.9 |
| | *MergeLock* | | 0.3 | 1.1 | 10.1 | 7.3 | 1.2 | 6.2 | 0.9 | 3.8 | ↓74.3 |
| | *MergeLock w/ Alig.* | | 0.3 | 1.3 | 6.2 | 5.2 | 2.1 | 21.4 | 1.4 | 5.4 | ↑1.6 |
| Cars | Normal | 64.7 | | 81.0 | 84.9 | 81.1 | 80.1 | 82.7 | 69.7 | 77.7 | |
| | PaRaMS | 1.5 | | 4.3 | 9.2 | 10.2 | 3.6 | 6.0 | 3.4 | 5.4 | ↓73.7 |
| | PaRaMS w/ Alig. | 63.8 | NA | 81.0 | 84.8 | 80.8 | 79.7 | 82.6 | 69.6 | 77.4 | ↑72.0 |
| | *MergeLock* | 0.3 | | 0.1 | 9.8 | 6.7 | 1.3 | 4.9 | 1.2 | 3.4 | ↓74.3 |
| | *MergeLock w/ Alig.* | 0.4 | | 1.5 | 6.4 | 5.4 | 2.2 | 26.6 | 1.2 | 6.2 | ↑2.8 |
| RESISC45 | Normal | 80.8 | 81.0 | | 90.8 | 92.3 | 92.2 | 95.0 | 81.4 | 87.6 | |
| | PaRaMS | 7.6 | 5.9 | | 17.4 | 17.2 | 11.0 | 13.0 | 9.7 | 11.6 | ↓76.0 |
| | PaRaMS w/ Alig. | 80.6 | 81.0 | NA | 90.9 | 92.2 | 92.4 | 95.0 | 81.2 | 87.6 | ↑76.0 |
| | *MergeLock* | 1.0 | 1.5 | | 4.8 | 7.3 | 2.1 | 6.7 | 1.5 | 3.5 | ↓84.1 |
| | *MergeLock w/ Alig.* | 1.2 | 1.2 | | 12.4 | 6.8 | 4.3 | 32.5 | 3.1 | 8.7 | ↑5.2 |
| EuroSAT | Normal | 84.4 | 84.9 | 90.8 | | 96.2 | 95.5 | 98.1 | 85.1 | 90.7 | |
| | PaRaMS | 11.3 | 11.2 | 18.5 | | 23.7 | 13.1 | 17.6 | 12.7 | 15.4 | ↓75.3 |
| | PaRaMS w/ Alig. | 84.3 | 84.8 | 90.9 | NA | 96.1 | 95.6 | 98.1 | 85.5 | 90.7 | ↑75.3 |
| | *MergeLock* | 9.9 | 10.4 | 5.5 | | 13.5 | 11.1 | 10.5 | 9.6 | 10.0 | ↓80.7 |
| | *MergeLock w/ Alig.* | 6.1 | 4.1 | 9.1 | | 12.3 | 8.1 | 32.2 | 6.9 | 11.2 | ↑1.2 |
| SVHN | Normal | 83.0 | 81.1 | 92.3 | 96.2 | | 93.6 | 96.1 | 80.4 | 88.9 | |
| | PaRaMS | 11.1 | 10.7 | 15.2 | 21.6 | | 17.2 | 18.0 | 13.7 | 15.3 | ↓73.6 |
| | PaRaMS w/ Alig. | 82.9 | 81.0 | 92.3 | 96.1 | NA | 93.6 | 96.0 | 80.4 | 88.9 | ↑73.6 |
| | *MergeLock* | 5.6 | 6.5 | 8.0 | 13.0 | | 6.4 | 12.9 | 6.8 | 8.4 | ↓80.5 |
| | *MergeLock w/ Alig.* | 5.0 | 5.1 | 6.5 | 12.0 | | 9.8 | 60.8 | 7.5 | 15.2 | ↑6.8 |
| GTSRB | Normal | 81.7 | 80.1 | 92.2 | 95.5 | 93.6 | | 95.6 | 80.4 | 88.4 | |
| | PaRaMS | 5.4 | 5.3 | 10.7 | 12.5 | 17.4 | | 10.9 | 7.3 | 9.9 | ↓78.5 |
| | PaRaMS w/ Alig. | 81.5 | 79.8 | 92.3 | 95.1 | 93.6 | NA | 95.5 | 80.1 | 88.2 | ↑78.3 |
| | *MergeLock* | 1.1 | 1.3 | 2.0 | 11.1 | 6.5 | | 7.9 | 1.9 | 4.5 | ↓83.9 |
| | *MergeLock w/ Alig.* | 1.7 | 2.3 | 4.6 | 8.5 | 11.5 | | 39.63 | 5.1 | 10.4 | ↑5.9 |
| MNIST | Normal | 84.8 | 82.7 | 95.0 | 98.1 | 96.1 | 95.6 | | 83.7 | 90.8 | |
| | PaRaMS | 6.7 | 6.1 | 12.2 | 14.9 | 16.5 | 10.0 | | 9.6 | 10.8 | ↓80 |
| | PaRaMS w/ Alig. | 84.7 | 82.6 | 95.2 | 98.2 | 96.1 | 95.8 | NA | 83.6 | 90.8 | ↑80 |
| | *MergeLock* | 6.0 | 5.0 | 6.6 | 7.8 | 13.6 | 5.3 | | 6.0 | 7.1 | ↓83.7 |
| | *MergeLock w/ Alig.* | 10.0 | 13.8 | 20.8 | 21.0 | 51.8 | 24.9 | | 15.6 | 22.5 | ↑15.4 |
| DTD | Normal | 67.9 | 69.7 | 81.4 | 85.1 | 80.4 | 80.4 | 83.7 | | 78.3 | |
| | PaRaMS | 3.1 | 4.3 | 7.8 | 11.0 | 12.9 | 5.5 | 9.0 | | 7.6 | ↓70.7 |
| | PaRaMS w/ Alig. | 67.2 | 69.5 | 81.1 | 85.2 | 80.7 | 80.1 | 83.5 | NA | 78.1 | ↑70.5 |
| | *MergeLock* | 0.8 | 1.0 | 1.7 | 11.4 | 8.1 | 1.9 | 7.1 | | 4.5 | ↓73.8 |
| | *MergeLock w/ Alig.* | 1.3 | 1.5 | 3.4 | 7.6 | 6.5 | 4.5 | 25.8 | | 7.2 | ↑2.7 |

often close to random guessing. This demonstrates the effectiveness of *MergeLock* in preventing successful model merging. (3) PaRaMS also reduces the accuracy of the merged models (the second row of each block), but the drop is generally less severe than that of *MergeLock*. For example, when merging a PaRaMS-protected SUN397 model with a normal Cars model, the accuracy is 1.4%, while merging a *MergeLock*-protected SUN397 model with a normal Cars model yields an accuracy of only 0.3%. This is because PaRaMS only applies diagonal scaling to self-attention layers, while our method combines three components: random perturbation, permutation, and diagonal scaling (in Sec. 3.4).

**Distance Analysis**. We further measure the Frobenius distance between protected and unprotected models to explain why our *MergeLock* method is more resistant to merging. Without loss of generality, we take the SUN397 dataset as an example. As shown in Fig. 2, for the Q&K and V&O matrices in each layer, we compute the Frobenius distance from the fine-tuned model to the pretrained model under three settings: the normal fine-tuned model (green), the PaRaMS-protected model (red), and the *MergeLock*-protected model (orange). We observe that the Frobenius distance for the normal fine-tuned model is very small—only 0.3 on average for the Q&K matrices—indicating that standard fine-tuning typically remains within the original loss basin. After applying perturbations, PaRaMS increases the average distance to 20.1, making models harder to merge. Notably, our *MergeLock* further increases the average distance to 270.3, effectively protecting the model and rendering it nearly impossible to merge. In Fig. 4 of the Appendix, we further verify that MergeLock disrupts the fundamental condition for effective model merging—linear mode connectivity.

### 4.2.2 EVALUATION OF ALIGNMENT ROBUSTNESS

In this subsection, we evaluate the robustness of *MergeLock* against alignment-based recovery attempts. Specifically, we consider the scenario where an adversary tries to reverse the protection by applying model alignment before merging. We investigate *whether our method can still effectively prevent merging under such circumstances*?

**Alignment Strategy**. In the model merging setting, attackers typically do not have access to the training data; therefore, we focus on data-free alignment strategies. The core idea of alignment is that the attacker applies transformation matrices ($R_1, R_2$ or $R_3, R_4$) to the parameters of the self-attention layers of two models—one protected (e.g., $W_{Q_1}, W_{K_1}, W_{V_1}$ or $W_{O_1}$) and one unprotected (e.g., $W_{Q_2}, W_{K_2}, W_{V_2}$ or $W_{O_2}$)—to perturb their weights such that the distance between them is minimized, i.e., they are brought into the same loss basin, thereby increasing the likelihood of successful merging. This leads to the following optimization problem for the $W_Q$ and $W_K$ projections:

$$\min_{R_1, R_2 \in \mathcal{R}} \left\| \begin{bmatrix} W_{Q_1}^\top & W_{Q_2}^\top \\ b_{Q_1} & b_{Q_2} \end{bmatrix} \begin{bmatrix} R_1 \\ -R_2 \end{bmatrix} \right\|_F^2 + \left\| \begin{bmatrix} W_{K_1}^\top & W_{K_2}^\top \\ b_{K_1} & b_{K_2} \end{bmatrix} \begin{bmatrix} R_1 \\ -R_2 \end{bmatrix} \right\|_F^2, \tag{10}$$

and similarly for the $W_V$ and $W_O$ projections:

$$\min_{R_3, R_4 \in \mathcal{R}} \left\| \begin{bmatrix} W_{V_1}^\top & W_{V_2}^\top \\ b_{V_1} & b_{V_2} \end{bmatrix} \begin{bmatrix} R_3 \\ -R_4 \end{bmatrix} \right\|_F^2 \left\| \begin{bmatrix} W_{O_1}^\top & W_{O_2}^\top \\ 0 & 0 \end{bmatrix} \begin{bmatrix} R_3 \\ -R_4 \end{bmatrix} \right\|_F^2. \tag{11}$$

As shown in Zhang et al. (2025), if $R$ is constrained to be a rotation matrix (i.e., orthogonal), the optimization in Eq. 10 admits a closed-form solution with the Kabsch-based algorithm (Kabsch, 1976; Umeyama, 1991): $R_1 = UV^\top, R_2 = \boldsymbol{I}$, where $\boldsymbol{I}$ denotes the identity matrix, and $U\Sigma V^\top$ is the singular value decomposition (SVD) of the matrix $W_{Q_1} W_{Q_2}^\top + W_{K_1} W_{K_2}^\top + b_{Q_1}^\top b_{Q_2} + b_{K_1}^\top b_{K_2}$. We implement this alignment process on *MergeLock* to test whether it can recover performance when merging our protected models. In addition, we also apply the Hungarian algorithm to PaRaMS-protected models for alignment, as done in the original PaRaMS paper (Wei et al., 2025). In Table 1, alignment-applied results are denoted with a "w/ Alig." suffix (third and fifth rows in each block).

**Performance Comparison**. As shown in Table 1, alignment improves the merged model's accuracy to some extent for both methods, confirming that partial parameter correspondence can be restored without data. However, the extent of recovery varies significantly between the two methods. For PaRaMS, many tasks regain more than 70% compared to the unaligned protected case, suggesting that its transformations are easier to invert when alignment is applied. For *MergeLock*, the improvement is much more modest, with typical gains of only a few percentage points. This shows that our method is robust against alignment-based attacks, as the residual mismatch in parameter space still severely impairs merging effectiveness. The Frobenius distances in Fig. 2 also confirm this point: after applying alignment, *MergeLock* w/ Alig. still exhibits large distances of 237.5 and 101.8 on Q&K and O&V matrices, respectively, far exceeding the original 20.1 and 9.6 of PaRaMS.

## 5 CONCLUSION AND FUTURE WORK

This paper proposes *MergeLock*, a method designed to address the growing safety concerns associated with model merging. *MergeLock* introduces two pairs of randomly sampled invertible matrices into the self-attention layers in Transformers, rendering the model unmergeable while preserving its original output behavior. Extensive experiments demonstrate that *MergeLock* significantly degrades the performance of merged models when a protected model is involved, and such degradation is difficult to recover through low-cost methods.

This work opens several promising directions for future research. First, given the widespread use of Transformers, this paper primarily focuses on protecting models based on the Transformer architecture; future work could extend to other network architectures. Second, the current approach provides a post-hoc protection strategy—protecting models after fine-tuning is completed—whereas future efforts could explore protecting models during the fine-tuning process itself. Finally, the proposed method could be applied to safeguard larger-scale models, such as large language models and multimodal large models, in model merging scenarios.

ETHICS STATEMENT

This work focuses on protecting the intellectual property of machine learning models from unauthorized merging. Our study does not involve human subjects, sensitive personal data, or ethically concerning datasets. All datasets used in this paper (SUN397, Stanford Cars, RESISC45, EuroSAT, SVHN, GTSRB, MNIST, and DTD) are publicly available benchmarks widely adopted in prior research. We acknowledge potential dual-use concerns: while our proposed method aims to safeguard developers against misuse, malicious actors might theoretically apply similar techniques to conceal harmful models. We emphasize that our intention is to promote secure model sharing and responsible AI development.

REPRODUCIBILITY STATEMENT

We have made every effort to ensure the reproducibility of our work. All datasets used in our experiments are publicly available, and detailed descriptions of model architectures, hyperparameters, and training procedures are provided in Sections 3–4 and the Appendix. Upon acceptance, we will release the full implementation, including the `MergeLock` transformations and evaluation scripts, on GitHub to facilitate independent verification.

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

APPENDIX CONTENTS

## A DATASETS

In this paper, we conduct experiments on eight vision classification datasets and eight text-to-text generation tasks to comprehensively evaluate our method.

**Vision Classification Tasks.** We utilize the following eight datasets for image classification, covering a diverse range of domains and complexities:

- **SUN397** is a large-scale scene classification dataset containing 108,754 images from 397 classes, with at least 100 images per class.
- **Stanford Cars (Cars)** contains 16,185 images from 196 car categories, split evenly (1:1) between training and test sets.
- **RESISC45** is a remote sensing image scene classification dataset with 31,500 images in 45 classes, each containing approximately 700 samples.
- **EuroSAT** consists of 27,000 geo-referenced satellite images from 10 classes.
- **SVHN** is a digit classification dataset from Google Street View house numbers, with 10 classes, 73,257 training images, 26,032 test images, and 531,131 additional samples.
- **GTSRB** contains over 50,000 images of 43 traffic sign classes.
- **MNIST** is a handwritten digit classification benchmark with 60,000 training and 10,000 test images evenly distributed across 10 classes.
- **DTD** is a texture classification dataset with 5,640 images across 47 classes, each containing approximately 120 samples.

**Text-to-Text Generation Tasks.** We also evaluate on eight GLUE benchmark tasks (Wang et al., 2018), including CoLA, MNLI, MRPC, QNLI, QQP, RTE, SST-2, and STSB, to verify the generality of our approach on language models.

## B DETAILS ABOUT PARAMS

PaRaMS (Wei et al., 2025) is a recent method designed to disrupt model merging by applying structured transformations to the parameters of fine-tuned models. It contains MLP protection and Self-attention protection.

**MLP Protection.** PaRaMS protects the two-layer MLP by permuting its hidden neurons to maximize the distance between the pre-trained and fine-tuned weights. Formally, given the pre-trained weights $\theta_{\text{pre}}^{\text{MLP}}$ and fine-tuned (protected) weights $\theta_{\text{ft}}^{\text{MLP}}$, the permutation $\eta_{\text{perm}}$ is chosen as:

$$\arg\max_{\eta_{\text{perm}}} \left\| \theta_{\text{pre}}^{\text{MLP}} - \eta_{\text{perm}}\left(\theta_{\text{ft}}^{\text{MLP}}\right) \right\|_F^2, \tag{12}$$

which is equivalent to:

$$\arg\min_{\eta_{\text{perm}}} \theta_{\text{pre}}^{\text{MLP}} \cdot \eta_{\text{perm}}\left(\theta_{\text{ft}}^{\text{MLP}}\right). \tag{13}$$

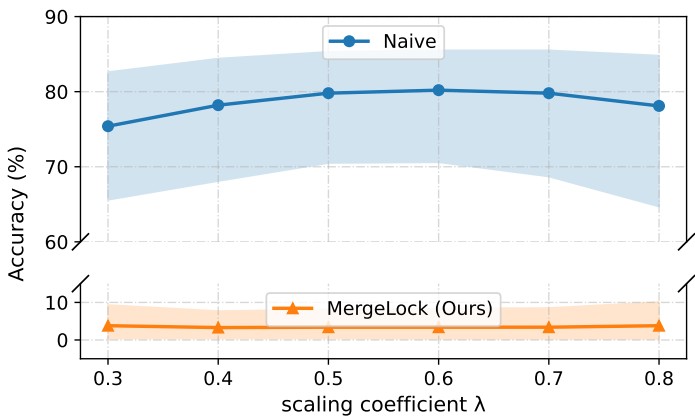

Figure 3: Average accuracy of SUN397 (fine-tuned on ViT-B/32) after being merged with each of seven other tasks individually using various scaling coefficients $\lambda$. The blue line represents unprotected merging, while the orange line shows the result when SUN397 is protected by our method. Accuracy values are averaged over the seven pairwise merging results.

For a two-layer MLP with first-layer weight $W^{\mathrm{mlp1}} \in \mathbb{R}^{d_h \times d_m}$ and second-layer weight $W^{\mathrm{mlp2}} \in \mathbb{R}^{d_m \times d_h}$, where $d_m$ is the hidden dimension and $d_h$ is the input/output dimension, the permutation $\eta_{\mathrm{perm}}()$ acts on the hidden neurons as:

$$\arg \min_{\eta_{\mathrm{perm}}=\{P_i\}_{i=1}^n} \sum_{i=1}^n \left\langle W_{\mathrm{pre}}^{(mlp1,i)},\, P_i\, W_{\mathrm{ft}}^{(mlp1,i)} \right\rangle_F + \left\langle W_{\mathrm{pre}}^{(mlp2,i)},\, W_{\mathrm{ft}}^{(mlp2,i)} P_i^\top \right\rangle_F \tag{14}$$

where $P_i$ is a one-hot permutation matrix for neuron $i$, and $\langle A, B \rangle_F$ denotes the Frobenius inner product $\mathrm{Tr}(A^\top B)$. This problem is a *linear assignment problem*, which PaRaMS solves exactly using the Hungarian algorithm to find the permutation $P_i$ that maximizes the mismatch between corresponding neurons. However, this step is inherently reversible: an adversary with access to the pre-trained model can resolve the same assignment problem with the objective $\max$ replaced by $\min$, directly recovering the permutation that minimizes the parameter distance, thus undoing the protection.

**Self-attention Protection.** In the self-attention layers, PaRaMS inserts two pairs of mutually-invertible matrices $(A, A^{-1})$ and $(B, B^{-1})$ into the $QK$ and $VO$ branches, respectively, in order to alter the attention computation while preserving functional equivalence for the fine-tuned task. Concretely, $Q$ and $K$ are multiplied by $A$ and $A^{-1}$, while $V$ and $O$ are multiplied by $B$ and $B^{-1}$. Importantly, PaRaMS chooses $A$ and $B$ as *diagonal scaling matrices*, which significantly limits the transformation's complexity. Since diagonal scaling preserves the axis-aligned structure of the parameter space, such transformations can be aligned with high accuracy by estimating per-dimension scaling factors, making this component relatively easy to reverse-engineer.

Overall, PaRaMS applies structured transformations to both MLP and self-attention layers to maximize parameter mismatch while preserving task performance. However, both components rely on reversible operations (permutations and diagonal scalings) that can be effectively countered by an informed adversary with access to the pre-trained model.

## C    SCALING COEFFICIENT ANALYSIS

Task Arithmetic (Ilharco et al., 2023) merges two fine-tuned models by linearly combining their parameter differences from a shared pre-trained model, scaled by a coefficient $\lambda$. Specifically, given two fine-tuned models $\theta_1$ and $\theta_2$ derived from a common pre-trained model $\theta_{\mathrm{pre}}$, the merged model

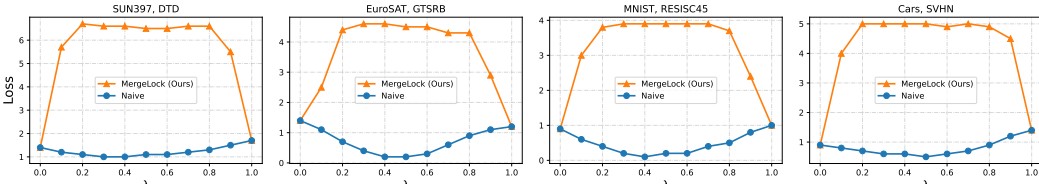

Figure 4: Linearly Mode Connectivity (LMC) curves between pairs of models across four dataset groups: SUN397/DTD, EuroSAT/GTSRB, MNIST/RESISC45, and Cars/SVHN. Blue lines indicate normal fine-tuned models, while orange lines denote cases where one model is made unmergeable. Unmergeable models exhibit significantly higher loss across interpolation, suggesting they escape the shared loss basin.

$\theta_{\text{merged}}$ is computed as:

$$\theta_{\text{merged}} = \theta_{\text{pre}} + \lambda \sum_{k=1}^{2} (\theta_k - \theta_{\text{pre}}), \tag{15}$$

where $\lambda > 0$ controls the relative contribution degree of the aggregated task vectors.

Considering that the performance of Task Arithmetic is influenced by the merging coefficient $\lambda$, we investigate how varying $\lambda$ affects the performance of our method. As shown in Fig. 3, when merging two models without protection, the choice of $\lambda$ leads to a performance fluctuation of nearly 5 percentage points, with the highest accuracy achieved around $\lambda = 0.6$. This sensitivity indicates that proper coefficient tuning is necessary to obtain optimal merging results in the unprotected setting. In contrast, when the SUN397 model is protected by our method, the merged model's accuracy remains consistently low (around 2–5%) across all tested $\lambda$ values, showing only negligible variation. This stability demonstrates that our protection mechanism effectively eliminates the benefits of coefficient tuning, making the merged model unusable regardless of $\lambda$. These results highlight that our approach not only degrades the merged model's performance but also neutralizes potential performance gains from hyperparameter optimization during merging.

## D  LINEARLY MODE CONNECTIVITY ANALYSIS

Linear Mode Connectivity (LMC) (Frankle et al., 2020) characterizes the geometry of the loss landscape between two models by linearly interpolating their parameters and measuring the resulting loss. It reveals whether the two models occupy the same loss basin, which is a key assumption underlying model merging techniques. If two models are in the same basin, the loss remains low along the interpolation path; otherwise, a significant loss barrier appears. More formally, given two models $\theta_1$ and $\theta_2$, LMC evaluates the loss along the linear path connecting them in parameter space:

$$\mathcal{L}(f(\theta_{\text{pre}} + \lambda(\theta_1 - \theta_{pre}) + (1 - \lambda)(\theta_2 - \theta_{pre})), \ \lambda \in [0, 1] \tag{16}$$

where $f(\cdot)$ is the model's prediction function, $\mathcal{L}(\cdot)$ is the task loss (e.g., cross-entropy), and $\theta_{\text{pre}}$ is the pretrained weight, and $\lambda$ controls the interpolation ratio.

As shown in Fig. 4, we evaluate LMC for four representative dataset pairs. The x-axis represents the interpolation coefficient $\lambda$, while the y-axis shows the corresponding loss value. The leftmost point ($\lambda = 0$) corresponds to model $\theta_2$, and the rightmost point ($\lambda = 1$) corresponds to model $\theta_1$. For each pair, we compare two scenarios: (1) **Normal (blue curves):** Both models are normally fine-tuned without protection. We observe low loss throughout the interpolation, indicating they share a common loss basin. (2) **Unmergeable (orange curves):** One model is normally fine-tuned, while the other is protected by our method to be unmergeable. We observe a pronounced loss spike across almost the entire interpolation path, indicating that the protected model has moved to a different loss basin. Therefore, LMC analysis reveals that our *MergeLock* method effectively enforces loss-basin isolation, breaking the assumption of basin compatibility that merging methods rely on.

Table 2: Classification accuracy (%) of ViT-L/14 models merged via Task Arithmetic. Avg. denotes the average of the merged models in each row; in MergeLock, Δ is the performance gap w/ vs. w/o the protection mechanism; in MergeLock w/ Alig., Δ is the gap w/ vs. w/o alignment.

| | | SUN397 | Cars | RESISC45 | EuroSAT | SVHN | GTSRB | MNIST | DTD | Avg. | Δ |
|---|---|---|---|---|---|---|---|---|---|---|---|
| SUN397 | Normal | NA | 86.3 | 88.1 | 89.3 | 89.3 | 89.3 | 85.8 | 82.1 | 87.1 | |
| | *MergeLock* | | 0.3 | 1.2 | 4.5 | 5.2 | 1.2 | 5.0 | 1.7 | 2.7 | ↓84.4 |
| | *MergeLock* w/ Alig. | | 0.4 | 1.6 | 10.3 | 8.6 | 1.3 | 4.9 | 11.5 | 5.5 | ↑2.8 |
| Cars | Normal | 86.3 | NA | 93.7 | 94.3 | 94.2 | 94.1 | 87.5 | 87.8 | 91.1 | |
| | *MergeLock* | 0.3 | | 1.4 | 4.5 | 4.9 | 1.3 | 5.1 | 1.4 | 2.7 | ↓88.4 |
| | *MergeLock* w/ Alig. | 0.3 | | 1.6 | 9.0 | 8.6 | 1.4 | 6.2 | 1.5 | 4.0 | ↑1.3 |
| RESISC45 | Normal | 88.1 | 93.7 | NA | 92.8 | 96.5 | 96.5 | 96.4 | 88.2 | 93.1 | |
| | *MergeLock* | 1.3 | 1.4 | | 5.5 | 7.8 | 2.0 | 6.9 | 2.4 | 3.9 | ↓89.2 |
| | *MergeLock* w/ Alig. | 1.4 | 1.5 | | 12.5 | 17.3 | 2.6 | 6.8 | 2.8 | 6.4 | ↑2.5 |
| EuroSAT | Normal | 89.3 | 94.3 | 92.8 | NA | 97.4 | 97.7 | 96.5 | 88.9 | 93.8 | |
| | *MergeLock* | 5.9 | 5.0 | 4.9 | | 7.6 | 4.4 | 8.2 | 6.5 | 6.0 | ↓87.8 |
| | *MergeLock* w/ Alig. | 5.4 | 7.4 | 10.4 | | 27.0 | 7.8 | 10.7 | 8.0 | 10.9 | ↑4.9 |
| SVHN | Normal | 89.3 | 94.2 | 96.5 | 97.4 | NA | 97.2 | 92.0 | 89.5 | 93.7 | |
| | *MergeLock* | 4.7 | 4.6 | 7.6 | 6.5 | | 5.8 | 9.7 | 6.2 | 6.4 | ↓87.3 |
| | *MergeLock* w/ Alig. | 6.6 | 5.9 | 10.2 | 18.9 | | 8.3 | 17.7 | 7.8 | 10.7 | ↑4.3 |
| GTSRB | Normal | 89.3 | 94.1 | 96.5 | 97.7 | 97.2 | NA | 96.7 | 89.4 | 94.4 | |
| | *MergeLock* | 1.2 | 1.5 | 2.8 | 4.5 | 6.8 | | 4.8 | 2.5 | 3.4 | ↓91.0 |
| | *MergeLock* w/ Alig. | 1.2 | 1.5 | 2.5 | 8.2 | 15.2 | | 5.4 | 2.7 | 5.2 | ↑1.8 |
| MNIST | Normal | 85.8 | 87.5 | 96.4 | 96.5 | 92.0 | 96.7 | NA | 89.4 | 92.0 | |
| | *MergeLock* | 5.5 | 5.2 | 9.7 | 8.7 | 5.9 | 7.5 | | 1.7 | 6.3 | ↓85.7 |
| | *MergeLock* w/ Alig. | 4.6 | 4.5 | 7.2 | 12.4 | 20.3 | 5.6 | | 5.6 | 8.6 | ↑2.3 |
| DTD | Normal | 82.1 | 87.8 | 88.2 | 88.9 | 89.5 | 89.4 | 89.4 | NA | 87.9 | |
| | *MergeLock* | 1.4 | 2.3 | 6.6 | 6.8 | 2.4 | 5.9 | 0.4 | | 3.7 | ↓84.2 |
| | *MergeLock* w/ Alig. | 1.6 | 1.3 | 2.9 | 9.2 | 11.3 | 2.7 | 6.1 | | 5.0 | ↑1.3 |

# E EXTENDED EXPERIMENTS ON MODEL SIZE, MERGING STRATEGIES, AND ARCHITECTURES

In the main text, we demonstrate the effectiveness of our proposed protection method, *MergeLock*, on ViT-B/32 models merged via Task Arithmetic (e.g., Tab. 1). In this section, to further evaluate the generality and robustness of our proposed protection method, we extend our experiments to (1) models with larger parameter sizes, (2) different model merging strategies beyond Task Arithmetic, and (3) different model architectures beyond vision Transformers.

**Experimental Setup.** We conduct experiments on the same eight vision classification datasets as in the main text for vision models, and on eight GLUE benchmark tasks for FlanT5. For each dataset, we fine-tune a pre-trained model (ViT-B/32, ViT-L/14, or FlanT5) to obtain a task-specific model. Our protection method is then applied to one of the fine-tuned models to make it unmergeable, while the other model remains unprotected. Specifically, we use ViT-L/14 to assess scalability with respect to model size, evaluate two additional merging algorithms—Ties-Merging (Yadav et al., 2023a) (conflict mitigation) and AdaMerging (Yang et al., 2024c) (data-driven adaptive merging)—and validate our approach on FlanT5, a Transformer-based encoder–decoder model for language tasks. In each experiment, we select one fine-tuned model as the target for protection and keep another model unprotected, following the same protocol as in the main experiments. We compare three settings: *Normal* (no protection), *MergeLock* (our protection), and *MergeLock w/ Alig.* (our protection followed by alignment attack). For vision models, accuracy is reported on the eight classification datasets; for FlanT5, accuracy is reported on the GLUE benchmark datasets. We also report the average accuracy (Avg.) and the difference (Δ) from the baseline (*normal*) setting.

**Impact of Model Size (ViT-L/14, Tab. 2).** When merging two unprotected ViT-L/14 models via Task Arithmetic, accuracies range from 85% to 94% across all datasets. With *MergeLock* protection applied, the merged model's performance drops sharply by 84–91 percentage points on average, rendering it unusable. Alignment recovery (*MergeLock w/ Alig.*) marginally improves performance by only 1–3%, confirming that larger model capacity does not weaken the protection effect.

**Impact on Tuning-free Merging Method (Ties-Merging, Tab. 3).** Ties-Merging (Yadav et al., 2023b) is a recently proposed approach designed to resolve interference when merging models. It first removes neurons with small magnitudes in the task vectors and further resolves parameter sign

Table 3: Classification accuracy (%) of ViT-B/32 models merged via Ties-Merging. Avg. denotes the average of the merged models in each row; in MergeLock, $\Delta$ is the performance gap w/ vs. w/o the protection mechanism; in MergeLock w/ Alig., $\Delta$ is the gap w/ vs. w/o alignment.

| | | SUN397 | Cars | RESISC45 | EuroSAT | SVHN | GTSRB | MNIST | DTD | Avg. | $\Delta$ |
|---|---|---|---|---|---|---|---|---|---|---|---|
| SUN397 | Normal | | 74.5 | 82.2 | 85.3 | 84.1 | 84.4 | 86.6 | 72.4 | 81.3 | |
| | *MergeLock* | NA | 0.4 | 1.7 | 4.7 | 5.8 | 1.1 | 5.7 | 1.6 | 3 | ↓78.3 |
| | *MergeLock* w/ Alig. | | 0.3 | 1.4 | 10.7 | 3.8 | 1.0 | 5.1 | 1.2 | 3.3 | ↑0.3 |
| Cars | Normal | 73.4 | | 83.0 | 85.3 | 84.4 | 84.5 | 86.7 | 73.7 | 81.5 | |
| | *MergeLock* | 0.4 | NA | 1.7 | 5.0 | 6.1 | 1.3 | 2.9 | 1.8 | 2.7 | ↓78.8 |
| | *MergeLock* w/ Alig. | 0.3 | | 1.6 | 10.8 | 3.8 | 1.4 | 5.2 | 1.6 | 3.5 | ↑0.8 |
| RESISC45 | Normal | 92.2 | 83.0 | | 92.1 | 92.1 | 92.7 | 95.4 | 81.3 | 89.8 | |
| | *MergeLock* | 1.7 | 1.8 | NA | 6.6 | 7.2 | 2.2 | 6.8 | 2.7 | 4.1 | ↓85.7 |
| | *MergeLock* w/ Alig. | 1.4 | 1.6 | | 6.3 | 5.1 | 2.6 | 6.3 | 2.7 | 3.7 | ↓0.4 |
| EuroSAT | Normal | 85.3 | 85.3 | 92.1 | | 93.4 | 94.6 | 97.7 | 84.4 | 90.4 | |
| | *MergeLock* | 4.6 | 4.7 | 6.3 | NA | 11.3 | 5.6 | 6.4 | 5.9 | 6.4 | ↓84 |
| | *MergeLock* w/ Alig. | 10.0 | 6.6 | 11.1 | | 12.1 | 11.3 | 10.1 | 6.6 | 9.6 | ↑3.2 |
| SVHN | Normal | 84.1 | 84.4 | 92.1 | 93.4 | | 94.7 | 96.7 | 81.9 | 89.6 | |
| | *MergeLock* | 5.0 | 5.1 | 6.4 | 10.0 | NA | 5.8 | 9.8 | 6.2 | 6.9 | ↓82.7 |
| | *MergeLock* w/ Alig. | 4.1 | 4.0 | 5.5 | 9.8 | | 5.3 | 9.0 | 5.3 | 6.1 | ↓0.8 |
| GTSRB | Normal | 84.4 | 84.5 | 92.7 | 94.6 | 94.7 | | 97.5 | 82.9 | 90.1 | |
| | *MergeLock* | 1.1 | 1.3 | 2.4 | 5.4 | 7.0 | NA | 3.2 | 2.2 | 3.2 | ↓86.9 |
| | *MergeLock* w/ Alig. | 1.6 | 1.7 | 2.7 | 6.3 | 5.8 | | 7.2 | 2.6 | 3.9 | ↑0.7 |
| MNIST | Normal | 86.6 | 86.7 | 95.4 | 97.7 | 96.7 | 97.5 | | 85.1 | 92.2 | |
| | *MergeLock* | 5.2 | 7.4 | 3.3 | 12.0 | 7.5 | 5.7 | NA | 7.8 | 6.9 | ↓85.3 |
| | *MergeLock* w/ Alig. | 5.1 | 5.2 | 6.9 | 10.5 | 9.2 | 5.6 | | 6.3 | 6.9 | ↑0 |
| DTD | Normal | 72.2 | 73.7 | 81.3 | 84.4 | 81.9 | 82.9 | 85.1 | | 80.2 | |
| | *MergeLock* | 1.6 | 1.9 | 2.8 | 6.0 | 7.7 | 2.3 | 5.4 | NA | 3.9 | ↓76.3 |
| | *MergeLock* w/ Alig. | 1.0 | 1.5 | 3.0 | 11.1 | 5.5 | 2.3 | 6.3 | | 4.3 | ↑0.4 |

conflicts, thereby reducing interference in model merging. In the unprotected setting, compared to Task Arithmetic (e.g., Tab. 1), Ties-Merging typically achieves higher accuracy when merging normally fine-tuned models. However, under our *MergeLock* protection, the average accuracy decreases by 76–87 percentage points, with most tasks falling below $10\%$. Alignment recovery remains minimal ($< 1\%$ in most cases), indicating that even advanced conflict-mitigation strategies such as Ties-Merging cannot bridge the large parameter-space gap introduced by our method. This highlights the robustness of our protection against stronger merging baselines.

**Impact on Tuning-based Merging Method (AdaMerging, Tab.4).** AdaMerging (Yang et al., 2024c) is a recent adaptive model merging method designed to overcome task interference without requiring additional training data. AdaMerging encourages the merged model to produce more confident predictions, thereby adaptively learning task-specific weighting without the need for ground-truth labels. This design makes AdaMerging both data-efficient and effective, often outperforming static baselines in unprotected scenarios. In our ViT-B/32 experiments, it achieves 79–93% accuracy when no protection is applied. However, when one of the models is protected with our *MergeLock* method, AdaMerging's entropy-driven optimization fails to compensate for the structural parameter transformation. As shown in Tab. 4, average accuracy drops sharply by 72–83 percentage points, with most tasks degrading close to random-guessing levels. Even with alignment-based recovery attacks, performance gains remain negligible (0–1%). These results indicate that the unsupervised entropy minimization strategy cannot bridge the loss-basin separation created by our protection. In contrast to its strong performance under normal settings, AdaMerging is rendered ineffective against our method, highlighting the robustness and generality of the proposed protection.

**Cross-Architecture Validation (Flan-T5, Tab. 5).** To further evaluate the generality of our method on a fundamentally different Transformer architecture, we conduct experiments on FlanT5, which adopts an encoder–decoder design. The encoder stack contains self-attention layers, while each decoder block includes both self-attention and encoder–decoder cross-attention modules. In this work, we apply the proposed protection (*MergeLock*) to all self-attention branches (in encoder and decoder) using the same invertible transformation design. Although the cross-attention modules are left unchanged in our current implementation, the results in Table 5 show that applying *MergeLock* to self-attention alone is sufficient to substantially degrade the merging performance, with alignment recovery remaining minimal. This indicates that self-attention is a particularly sensitive and effective locus for protection, and suggests that extending the modification to cross-attention

Table 4: Classification accuracy (%) of ViT-B/32 models merged via AdaMerging. Avg. denotes the average of the merged models in each row; in MergeLock, $\Delta$ is the performance gap w/ vs. w/o the protection mechanism; in MergeLock w/ Alig., $\Delta$ is the gap w/ vs. w/o alignment.

| | | SUN397 | Cars | RESISC45 | EuroSAT | SVHN | GTSRB | MNIST | DTD | Avg. | $\Delta$ |
|---|---|---|---|---|---|---|---|---|---|---|---|
| SUN397 | Normal | | 71.7 | 79.8 | 84.0 | 81.3 | 81.8 | 85.0 | 69.2 | 79.0 | |
| | *MergeLock* | NA | 0.4 | 0.9 | 5.9 | 8.6 | 1.2 | 4.7 | 1.2 | 3.3 | ↓75.7 |
| | *MergeLock* **w/ Alig.** | | 0.4 | 1.5 | 4.7 | 4.4 | 2.8 | 5.0 | 1.4 | 2.9 | ↓0.4 |
| Cars | Normal | 71.7 | | 80.5 | 84.0 | 81.8 | 81.5 | 84.9 | 70.0 | 79.2 | |
| | *MergeLock* | 0.4 | NA | 1.1 | 6.2 | 7.8 | 1.4 | 5.9 | 1.1 | 3.4 | ↓75.8 |
| | *MergeLock* **w/ Alig.** | 0.4 | | 1.5 | 4.8 | 4.4 | 1.8 | 6.6 | 1.6 | 3.0 | ↓0.4 |
| RESISC45 | Normal | 79.9 | 80.5 | | 91.2 | 89.1 | 89.4 | 93.2 | 77.1 | 85.8 | |
| | *MergeLock* | 0.8 | 1.0 | NA | 6.9 | 8.5 | 1.8 | 6.5 | 1.2 | 3.8 | ↓82.0 |
| | *MergeLock* **w/ Alig.** | 1.4 | 1.2 | | 5.9 | 6.0 | 3.9 | 6.3 | 2.5 | 3.9 | ↑0.1 |
| EuroSAT | Normal | 84.0 | 84.0 | 91.1 | | 92.1 | 93.0 | 97.2 | 81.8 | 89.0 | |
| | *MergeLock* | 5.9 | 6.1 | 6.9 | NA | 11.7 | 6.8 | 11.9 | 6.6 | 8.0 | ↓81.0 |
| | *MergeLock* **w/ Alig.** | 4.6 | 4.8 | 5.9 | | 9.3 | 6.1 | 10.1 | 5.6 | 6.6 | ↓1.4 |
| SVHN | Normal | 81.4 | 81.7 | 89.1 | 92.2 | | 91.9 | 95.4 | 78.8 | 87.2 | |
| | *MergeLock* | 7.9 | 7.2 | 8.1 | 11.0 | NA | 8.6 | 14.2 | 9.6 | 9.5 | ↓77.7 |
| | *MergeLock* **w/ Alig.** | 4.5 | 4.4 | 6.1 | 9.3 | | 6.8 | 9.8 | 6.0 | 6.7 | ↓2.8 |
| GTSRB | Normal | 81.8 | 81.5 | 89.4 | 92.9 | 91.8 | | 95.0 | 79.4 | 87.4 | |
| | *MergeLock* | 1.2 | 1.3 | 1.7 | 6.9 | 9.2 | NA | 6.4 | 2.0 | 4.1 | ↓83.3 |
| | *MergeLock* **w/ Alig.** | 2.8 | 1.8 | 4.0 | 7.1 | 7.5 | | 6.5 | 3.6 | 4.8 | ↑0.7 |
| MNIST | Normal | 85.0 | 84.9 | 93.3 | 97.2 | 95.4 | 95.1 | | 82.2 | 90.4 | |
| | *MergeLock* | 5.8 | 5.9 | 6.5 | 13.5 | 15.1 | 5.8 | NA | 7.2 | 8.5 | ↓81.9 |
| | *MergeLock* **w/ Alig.** | 5.2 | 5.3 | 6.0 | 8.5 | 9.8 | 6.5 | | 6.3 | 6.8 | ↓1.7 |
| DTD | Normal | 69.2 | 70.6 | 77.2 | 81.7 | 78.6 | 79.4 | 82.3 | | 76.9 | |
| | *MergeLock* | 1.2 | 1.2 | 1.4 | 6.7 | 10.0 | 2.1 | 6.8 | NA | 4.2 | ↓72.7 |
| | *MergeLock* **w/ Alig.** | 1.4 | 1.8 | 2.5 | 5.7 | 6.1 | 3.9 | 5.4 | | 3.8 | ↓0.4 |

would likely further strengthen the defense. These findings confirm that our approach generalizes across architectures, being effective for both encoder-only (ViT-B/32 and ViT-L/14) and encoder–decoder (FlanT5) models.

**Summary.** In this section, to validate the robustness and generality of our proposed *MergeLock* method, we conduct extensive experiments across three dimensions: (1) model size (ViT-B/32 vs. ViT-L/14), (2) merging strategy (Task Arithmetic vs. Ties-Merging vs. AdaMerging), and (3) model architecture (ViT vs. FlanT5). In all scenarios, our method consistently degrades the performance of merged models to near-random levels, with alignment-based recovery yielding only marginal improvements. These results underscore the effectiveness of our approach in protecting fine-tuned models from unauthorized merging across diverse settings, highlighting its *model-size independence, merge-strategy independence, and architecture independence*.

Table 5: Experimental results of merging Flan-T5-base models on all eight tasks via Task Arithmetic. Avg. denotes the average of the merged models in each row; in MergeLock, $\Delta$ is the performance gap w/ vs. w/o the protection mechanism; in MergeLock w/ Alig., $\Delta$ is the gap w/ vs. w/o alignment.

| | | COLA | MNLI | MRPC | QNLI | QQP | RTE | SST2 | STSB | Avg. | $\Delta$ |
|---|---|---|---|---|---|---|---|---|---|---|---|
| | Normal | | 77.2 | 79.3 | 80.5 | 78.3 | 76.0 | 82.0 | 79.5 | 79.0 | |
| COLA | *MergeLock* | NA | 0.0 | 0.0 | 0.0 | 0.0 | 0.0 | 0.0 | 0.0 | 0.0 | ↓79.0 |
| | *MergeLock* **w/ Alig.** | | 14.2 | 9.8 | 14.7 | 9.1 | 11.5 | 13.4 | 0.0 | 10.3 | ↑10.3 |
| | Normal | 77.2 | | 82.8 | 86.3 | 83.6 | 80.8 | 88.1 | 84.3 | 83.3 | |
| MNLI | *MergeLock* | 0.0 | NA | 0.0 | 0.0 | 0.0 | 0.0 | 0.0 | 0.0 | 0.0 | ↓83.3 |
| | *MergeLock* **w/ Alig.** | 19.0 | | 0.3 | 0.5 | 0.9 | 1.7 | 2.9 | 0.0 | 3.6 | ↑3.6 |
| | Normal | 79.3 | 82.8 | | 87.9 | 84.5 | 78.6 | 89.3 | 87.2 | 84.2 | |
| MRPC | *MergeLock* | 0.0 | 0.0 | NA | 0.0 | 0.0 | 0.0 | 0.0 | 0.0 | 0.0 | ↓84.2 |
| | *MergeLock* **w/ Alig.** | 16.4 | 0.2 | | 0.3 | 6.8 | 3.0 | 1.2 | 0.0 | 3.9 | ↑3.9 |
| | Normal | 80.5 | 86.3 | 87.9 | | 87.8 | 84.2 | 91.8 | 88.8 | 86.8 | |
| QNLI | *MergeLock* | 0.0 | 0.0 | 0.0 | NA | 0.0 | 0.0 | 0.0 | 0.8 | 0.1 | ↓86.7 |
| | *MergeLock* **w/ Alig.** | 20.6 | 0.2 | 0.7 | | 4.6 | 5.0 | 1.4 | 1.4 | 4.8 | ↑4.7 |
| | Normal | 78.3 | 83.6 | 84.5 | 87.8 | | 82.4 | 89.5 | 86.3 | 84.6 | |
| QQP | *MergeLock* | 0.0 | 0.0 | 0.0 | 0.0 | NA | 0.0 | 0.0 | 0.0 | 0.0 | ↓84.6 |
| | *MergeLock* **w/ Alig.** | 13.5 | 1.4 | 5.8 | 5.0 | | 7.9 | 4.4 | 1.6 | 5.7 | ↑5.7 |
| | Normal | 76.0 | 80.8 | 78.6 | 84.2 | 82.4 | | 86.8 | 82.8 | 81.7 | |
| RTE | *MergeLock* | 0.0 | 0.0 | 0.0 | 0.0 | 0.0 | NA | 0.0 | 0.0 | 0.0 | ↓81.7 |
| | *MergeLock* **w/ Alig.** | 20.8 | 1.4 | 2.4 | 3.5 | 5.2 | | 3.3 | 0.5 | 5.3 | ↑5.3 |
| | Normal | 82.0 | 88.1 | 89.3 | 91.8 | 89.5 | 86.8 | | 90.8 | 88.3 | |
| SST2 | *MergeLock* | 0.0 | 0.0 | 0.0 | 0.0 | 0.0 | 0.0 | NA | 0.0 | 0.0 | ↓88.3 |
| | *MergeLock* **w/ Alig.** | 18.3 | 2.1 | 1.6 | 1.8 | 3.4 | 4.2 | | 3.9 | 5.0 | ↑5.0 |
| | Normal | 79.5 | 84.3 | 87.2 | 88.8 | 86.3 | 82.8 | 90.8 | | 85.7 | |
| STSB | *MergeLock* | 0.0 | 0.0 | 0.0 | 0.0 | 0.0 | 0.0 | 0.0 | NA | 0.0 | ↓85.7 |
| | *MergeLock* **w/ Alig.** | 3.8 | 0.1 | 0.3 | 1.1 | 2.0 | 2.7 | 2.2 | | 1.7 | ↑1.7 |

