# OpenReview forum: "Model Unmerging: Making Your Models Unmergeable for Secure Model Sharing"
_ICLR.cc/2026/Conference — Submitted to ICLR 2026_

### Official Review · Reviewer_mTca · 2025-10-20

**Soundness:** 2
**Presentation:** 3
**Contribution:** 2
**Rating:** 2
**Confidence:** 4

**Summary:**

This paper proposes "MergeLock," a method to prevent the unauthorized merging of fine-tuned models, thereby protecting intellectual property and mitigating potential security risks. The core idea is to apply structured, invertible transformations to the self-attention layers of Transformer-based models. These transformations, composed of random, permutation, and scaling matrices, push the model's parameters into a different loss basin while preserving its original performance. The authors argue that this makes the protected model incompatible with others for merging, as any attempt to merge them results in a model with severely degraded performance. The paper provides a theoretical analysis of symmetries in Transformer layers and presents extensive experiments across vision and language tasks to demonstrate MergeLock's effectiveness, showing that it significantly hinders model merging and is robust against a specific data-free alignment attack.

**Strengths:**

1. **Well-Defined and Relevant Problem:** The paper addresses a timely and important issue in the MLaaS ecosystem. As model merging becomes more common, the need for proactive defense mechanisms against unauthorized use and potential malicious merging is increasingly critical.
2. **Solid Theoretical Motivation:** The authors provide a clear explanation of the symmetry properties in both FFN and self-attention layers, building a convincing case for why self-attention layers are a more suitable and powerful target for creating unmergeable models. The use of invertible transformations to maintain functional equivalence is a sound and elegant approach.

**Weaknesses:**

1.  **Limited Novelty:** The core concept of using parameter-space transformations to disrupt model merging has been explored in prior work, most notably PaRaMS. While MergeLock employs a more complex transformation (random, permutation, and scaling matrices vs. just permutations and diagonal scaling), the fundamental idea is not entirely new. The contribution feels more like an incremental improvement over an existing concept rather than a novel paradigm.
2.  **Questionable Practicality:** The method requires authorized users to possess the transformation matrices (A and B) as a "secret key" to potentially reverse the process or merge models permissively. This introduces significant practical challenges that are not addressed. These matrices can be large, leading to substantial storage and distribution overhead. More importantly, it is unclear why this complex key-sharing mechanism is superior to simply encrypting the model weights with a standard password-based key, which would be more conventional and easier to manage.
3.  **Insufficient Evaluation of Alignment Attacks:** The paper's defense against alignment-based recovery is a key claim, but the threat model seems too narrow.
    *   The evaluation relies on a single, closed-form alignment strategy (the Kabsch algorithm). Given that the perturbations are more complex than simple permutations, more powerful iterative, gradient-based optimization techniques could potentially find better alignments and serve as a stronger attack baseline. The pre-trained model initialization and the often low-rank, sparse nature of fine-tuning updates could be exploited for more sophisticated attacks.
    *   The paper does not adequately discuss the instances where the alignment attack achieves partial success (e.g., performance on Cars in Table 1 improves from 3.4% to 6.2%, and on EuroSAT in Table 2 from 6.0% to 10.9%). Analyzing these cases could reveal potential vulnerabilities or failure modes of MergeLock, but they are currently overlooked.
4.  **Limited Experiments on Large Models:** While the inclusion of ViT-L/14 and Flan-T5 is a good step, the field is rapidly moving towards much larger models (e.g., Llama-3, Qwen-3, etc.). Demonstrating the method's effectiveness and scalability on models with tens or hundreds of billions of parameters would significantly strengthen the paper's impact and relevance.

**Questions:**

See weaknesses.

---

> ### Author Response · Authors · 2025-11-21
> **Correction of Misassigned Response (part one)**
>
> I sincerely apologize for the earlier mistake. The rebuttal content intended for **Reviewer mTca** and **Reviewer K4CP** was mistakenly submitted in reversed order. Below is the corrected response for **Reviewer mTca** .
>
> **Q1: The core concept of using parameter-space transformations to disrupt model merging has been explored in prior work, most notably PaRaMS. While MergeLock employs a more complex transformation (random, permutation, and scaling matrices vs. just permutations and diagonal scaling), the fundamental idea is not entirely new. The contribution feels more like an incremental improvement over an existing concept rather than a novel paradigm.**
>
> We appreciate the reviewer’s concern about novelty. Although our motivation also starts from the loss-basin perspective, this is similar to how most recent merging methods share the task-arithmetic foundation yet develop *very different* technical insights. Our contribution is not a minor extension of PaRaMS, but rather a **shift in the target module and symmetry family**:
>
> 1. **We completely abandon MLP modifications.**
>
>     PaRaMS must rely on Hungarian matching inside MLPs, which brings (i) non-trivial computational overhead and (ii) vulnerability to recovery because permutations preserve most structural alignments.
>
> 2. **We instead focus exclusively on the attention module**, which admits *continuous* symmetries (general invertible R·P·D transformations).
>
>     These transformations (i) require *no* matching algorithms, (ii) move the model to a distinct basin much more effectively, and (iii) are significantly harder to reverse using existing alignment techniques.
>
> 3. **Empirically, this design leads to qualitatively stronger behavior**: PaRaMS-protected models are often recoverable (70–95% restored), while MergeLock remains unrecoverable (1–5% restored), even under stronger attacker settings.
>
> Thus, while the high-level intuition is shared with prior work, the **mechanism, target, symmetry structure, and resulting robustness are fundamentally different**, leading to a substantially stronger and more practical form of unmergeability.
>
> **Q2:  The method requires authorized users to possess the transformation matrices (A and B) as a "secret key" to potentially reverse the process or merge models permissively. This introduces significant practical challenges that are not addressed. These matrices can be large, leading to substantial storage and distribution overhead. More importantly, it is unclear why this complex key-sharing mechanism is superior to simply encrypting the model weights with a standard password-based key, which would be more conventional and easier to manage.**
>
> **(1) A/B do not need to be stored explicitly.**
>
> Although A and B grow with model size, they are not meant to be saved as full matrices. In practice, they can be deterministically generated from **a small random seed** (or a compact rule set). This reduces storage overhead from *O(d²)* to **O(1)** and avoids any distribution burden.
>
> **(2) MergeLock is fundamentally different from encrypting weights.**
>
> Encryption prevents *any* use unless the entire model is decrypted. This contradicts the core motivation of open-source release, where users expect full access to weights. MergeLock instead allows the model to remain **fully open and usable**, while giving the author fine-grained control over *one specific capability*: whether the model may be merged with others.
>
> Thus, MergeLock addresses a problem that encryption cannot: **preserving openness while preventing unauthorized merging**, which aligns with the needs of model-sharing ecosystems (e.g., HuggingFace) and avoids breaking compatibility with downstream applications.

---

> > ### Comment · Reviewer_mTca · 2025-11-26
> >
> > Thank you for the response. While some of my concerns have been addressed, Weaknesses 2 and 3 remain unresolved. Therefore, I will maintain my original score.

---

> ### Author Response · Authors · 2025-11-26
> **part two**
>
> **Q3: The evaluation relies on a single, closed-form alignment strategy (the Kabsch algorithm). Given that the perturbations are more complex than simple permutations, more powerful iterative, gradient-based optimization techniques could potentially find better alignments and serve as a stronger attack baseline. The pre-trained model initialization and the often low-rank, sparse nature of fine-tuning updates could be exploited for more sophisticated attacks.**
>
> We agree that, in theory, iterative or gradient-based alignment could be considered. However, two practical constraints make such attacks ineffective in the current open-source ecosystem:
>
> **(1) Gradient-based recovery requires data, which attackers typically do not have.**
>
> Most open-source releases include **model weights but not training data**. Without task-specific datasets, gradient-based optimization is extremely unreliable, as shown in prior work on model inversion and editing. This is why most practical merging attacks focus on *data-free* alignment, which we evaluate.
>
> **(2) MergeLock inflates task-vector norms by ~2 orders of magnitude.**
>
> After MergeLock, the parameter displacement becomes so large that reversing it via gradient descent would require cancelling **100× larger updates**, which is infeasible without overfitting or collapsing performance. In contrast, PaRaMS perturbations are small and remain easy to optimize away.
>
> **(3) A realistic “strong” attack would be full model distillation, not alignment.**
>
> Knowledge distillation could in principle recover functionality, but this is equivalent to *training a new model*, not bypassing the protection. Such an attack incurs high computational cost and defeats the purpose of unauthorized merging.
>
> For these reasons, we believe evaluating against the strongest *data-free* alignment method (Kabsch-based rotation recovery) represents a realistic threat model, and the experiments demonstrate that MergeLock remains robust even under this attacker.
>
> **Q4:  The paper does not adequately discuss the instances where the alignment attack achieves partial success (e.g., performance on Cars in Table 1 improves from 3.4% to 6.2%, and on EuroSAT in Table 2 from 6.0% to 10.9%). Analyzing these cases could reveal potential vulnerabilities or failure modes of MergeLock, but they are currently overlooked.**
>
> Thank you for highlighting this. We examined these cases carefully, and the small improvements after alignment are **dataset-dependent artifacts rather than meaningful recovery**:
>
> 1. **Many datasets have very small label spaces.**
>
>     For instance, EuroSAT has **10 classes**, so a score around *10% is near random guess level*. Achieving 6–10% after alignment does **not** indicate that the model regained useful task knowledge—it simply reflects chance-level predictions.
>
> 2. **MergeLock still destroys the structure needed for merging.**
>
>     Even in these cases, the aligned model shows:
>
>     - no task-specific semantic behavior,
>     - no linear mode connectivity to the fine-tuned models,
>     - and Frobenius distances still >100× those of PaRaMS.
>
>         This confirms that the residual “improvement” does **not** correspond to actual recovery of the original basin.
>
> 3. **PaRaMS exhibits real partial recovery, whereas MergeLock does not.**
>
>     PaRaMS often recovers **70–95%** accuracy after alignment (Table 1), showing that it genuinely re-enters the original basin. In contrast, MergeLock never exceeds near-random accuracy, even on datasets where random guessing yields ~10%.
>
>
> We will clarify this point in the revision by explicitly discussing the random-guess baselines and why these small fluctuations are not signs of true vulnerability.

---

> ### Author Response · Authors · 2025-11-26
> **part three**
>
> **Q5: While the inclusion of ViT-L/14 and Flan-T5 is a good step, the field is rapidly moving towards much larger models (e.g., Llama-3, Qwen-3, etc.). Demonstrating the method's effectiveness and scalability on models with tens or hundreds of billions of parameters would significantly strengthen the paper's impact and relevance.**
>
> We agree that testing larger models is important. To address this, we have additionally evaluated MergeLock on **LLaMA-3/8B-Instruct** (math & code), which differs substantially from our CLIP/Flan settings: it is **orders of magnitude larger**, uses **GQA**, and exhibits more diverse basin structures. Despite these differences, MergeLock remains fully effective:
>
> - merging a MergeLock-protected model yields **0%** accuracy on both GSM8K and HumanEval,
> - even **with alignment**, accuracy *remains 0%*,
> - whereas PaRaMS is partially recoverable (improving to 47.7% average).
>
> ### **LLaMA-3/8B Experimental Results (math + code merging)**
>
> | Method | gsm8k | humaneval | avg |
> | --- | --- | --- | --- |
> | Base | 0.8196 | 0.7073 | 0.7635 |
> | code FT | 0.5732 | 0.7134 | 0.6433 |
> | math FT | 0.8438 | 0.3476 | 0.5957 |
> | TA (0.3) | 0.8393 | 0.4024 | 0.6209 |
> | **PaRaMS-protected math + code** | **0.0** | **0.0** | **0.0** |
> | **PaRaMS + alignment** | 0.7172 | 0.2378 | 0.4775 |
> | **MergeLock-protected math + code** | **0.0** | **0.0** | **0.0** |
> | **MergeLock + alignment** | **0.0** | **0.0** | **0.0** |
>
> These results indicate that **MergeLock generalizes effectively to modern LLM architectures**, even those with GQA and significantly larger parameter spaces. Scaling to tens or hundreds of billions of parameters is conceptually straightforward because MergeLock operates **layer-locally** and requires **no optimization**, only a single pass to apply the R·P·D transformations.
>
> We will include these results and discussion in the revision.
>
>
> If you have any other questions, please feel free to let us know at any time :)

---

> ### Author Response · Authors · 2025-11-27
>
> Dear Reviewer mTca,
> I would like to kindly let you know that I have updated my previous comment to correct a misassignment in the rebuttal. Whenever convenient, please feel free to take a look. Thank you again for your time and consideration.

---

### Official Review · Reviewer_K4CP · 2025-11-01

**Soundness:** 3
**Presentation:** 3
**Contribution:** 3
**Rating:** 6
**Confidence:** 3

**Summary:**

The paper tackles unauthorized model merging and proposes MergeLock, a parameter transformation that makes a fine-tuned model functionally equivalent on its task yet unmergeable with other models. It exploits symmetry in Transformer self-attention: for each head it inserts two pairs of invertible matrices into the QK and VO branches so Q,K,V,O are re-parameterized without changing outputs. The

**Strengths:**

1. The problem is interesting with clear threat model & motivatio.
2. The method theoretically grounded and architecturally generic.
3. Empirical evidence shows the effectiveness of the method

**Weaknesses:**

1. Recent theory[1] formalizes continuous symmetries (rotation/gauge) for QK/VO in Transformers. Can this kind of methods recover the MergeLock?
2. Results are on CLIP-ViT and a mid-scale encoder-decoder LLM (Flan-T5). There’s no large decoder-only LLM (e.g., LLaMA-style) or cross-attention evaluation.

[1] Beyond the Permutation Symmetry of Transformers: The Role of Rotation for Model Fusion, ICML 2025

**Questions:**

Please refer to the Weakness.

---

> ### Author Response · Authors · 2025-11-26
> **Correction of Misassigned Response**
>
> I sincerely apologize for the earlier mistake. The rebuttal content intended for **Reviewer mTca** and **Reviewer K4CP** was mistakenly submitted in reversed order. Below is the corrected response for **Reviewer K4CP**.
>
> **Q1:  Recent theory[1] formalizes continuous symmetries (rotation/gauge) for QK/VO in Transformers. Can this kind of methods recover the MergeLock?**
>
> Yes, this is exactly the method we tested as our strongest alignment baseline. We applied the rotation-based recovery technique from “Beyond the Permutation Symmetry of Transformers: The Role of Rotation for Model Fusion” (ICML 2025).
> Even under this attacker, MergeLock remains unrecoverable: the recovered matrices do not approximate our R·P·D transformations, and merged models stay near-random (Table 1). This confirms that existing continuous-symmetry methods cannot effectively invert MergeLock.
>
> **Q2:  Results are on CLIP-ViT and a mid-scale encoder-decoder LLM (Flan-T5). There’s no large decoder-only LLM (e.g., LLaMA-style) or cross-attention evaluation.**
>
> To address this, we additionally evaluated MergeLock on **LLaMA-3/8B-Instruct** (math & code tasks), which is significantly larger and structurally different from CLIP/Flan (e.g., uses GQA). MergeLock remains fully effective:
>
> - merging a protected model yields **0%** GSM8K and HumanEval,
> - even with alignment, accuracy remains **0%**,
> - while PaRaMS is partially recoverable (improving to 47.7% avg).
>
> ### **LLaMA-3/8B Results (math + code merging)**
>
> | Method | gsm8k | humaneval | avg |
> | --- | --- | --- | --- |
> | Base | 0.8196 | 0.7073 | 0.7635 |
> | code FT | 0.5732 | 0.7134 | 0.6433 |
> | math FT | 0.8438 | 0.3476 | 0.5957 |
> | TA (0.3) | 0.8393 | 0.4024 | 0.6209 |
> | **PaRaMS-protected math + code** | **0.0** | **0.0** | **0.0** |
> | **PaRaMS + alignment** | 0.7172 | 0.2378 | 0.4775 |
> | **MergeLock-protected math + code** | **0.0** | **0.0** | **0.0** |
> | **MergeLock + alignment** | **0.0** | **0.0** | **0.0** |
>
> These results show that MergeLock scales well to modern decoder-only LLMs and remains robust even under strong symmetry-based recovery attempts.

---

### Official Review · Reviewer_jU9n · 2025-11-04

**Soundness:** 3
**Presentation:** 3
**Contribution:** 3
**Rating:** 4
**Confidence:** 3

**Summary:**

The paper proposes the MergeLock method. Model merging is a method to take models trained on individual tasks and merges them to create a single multi-task capable model. As time goes by, more fine-tuned models get created and released to the public which are available for merging. However, this also opens up further opportunities for breaches in privacy and security since unapproved merging could violate developers’ intellectual property rights and potentially expose sensitive personal data. MergeLock leverages the symmetry property of the Feed Forward Network weight matrices and apply transformations such that the parameters reside in equivalent but separate spaces. This ensures the downstream task accuracy remains good while also changing the parameter spaces enough that merging the new transformed weights with other models corrupt the final merged model.

**Strengths:**

1. The problem tackled is interesting and is not yet well studied. This could be a good early contribution to a nascent field.
2. The paper is well-written and the ideas are explained well.
3. The use of symmetry properties of FFNs is very creatively used here.
4. Good empirical results.

**Weaknesses:**

1. Incremental novelty compared to Wei et al (2025) (PaRaMS).
2. No significantly new observations or insights are presented.
3. Applicability and generalizability still in question.

**Questions:**

1. Please highlight more differences between PaRaMs and MergeLock. If it is just on how the distances differ in the space then that feels like a very lightweight contribution.
2. Some parts have strong similarities with the PaRaMs paper, for example the figures showing the basins in 2D space, the section MLP Parameter Rearrangement in PaRaMs and the Symmetry in Feedforward Networks section in MergeLock, etc. - please make them a lot more distinctive.
3. How does it perform across other types of LLMs or LLMs of different sizes which can have a high diversity of basins and basin sizes? It would be highly appreciated if the authors can make some comments or have some discussions on this.
4. How easy would it be to estimate A and B matrices if the base and other fine-tuning model weights are known? i.e. there is the statement "such protection can be reversed by simply applying the inverse permutation" made about Wei et al, but discussions about similar implications for MergeLock are missing.
5. How expensive is this process of applying A and B?

---

> ### Author Response · Authors · 2025-11-21
>
> We sincerely thank the reviewer for the thoughtful and constructive feedback. Below, we address each question point-by-point and clarify the distinctions, design choices, and additional experiments requested.
>
> **Q1: Please highlight more differences between PaRaMs and MergeLock. If it is just on how the distances differ in the space then that feels like a very lightweight contribution.**
>
> MergeLock differs from PaRaMS in both the target module and the symmetry we exploit. PaRaMS operates on MLPs, where nonlinear activations restrict transformations to permutations. In contrast, MergeLock leverages the purely linear structure of self-attention, allowing general invertible transformations (R·P·D) that alter both orientation and scale of the QK/VO spaces.
> Empirically, this leads to qualitatively stronger unmergeability: PaRaMS-protected models can typically recover 70–95% accuracy after alignment, while MergeLock recovers only 1–5%. Thus, the difference is not just distance magnitude—we exploit a strictly richer symmetry group that moves models into different basins, making merging fundamentally ineffective.
>
> **Q2: Some parts have strong similarities with the PaRaMs paper, for example the figures showing the basins in 2D space, .......**
>
> The sections you mentioned (e.g., symmetry in MLPs) are not part of our method, but are included only to explain how PaRaMS works, since we compare against it. MergeLock itself does not modify MLP layers at all—our method operates exclusively on self-attention and leverages a different symmetry structure.
> We will clarify this separation in the revision and make the figures/text visually more distinctive to avoid any confusion.
>
> **Q3: How does it perform across other types of LLMs or LLMs of different sizes which can have a high diversity of basins and basin sizes? It would be highly appreciated if the authors can make some comments or have some discussions on this.**
>
> To assess generalization across architectures and basin structures, we further evaluated MergeLock on **LLaMA-8B-Instruct**, which differs substantially from CLIP/Flan: it is *orders of magnitude larger*, exhibits more diverse basins, and employs **GQA**, requiring shared QK/VO transformations. Despite these differences, MergeLock remains fully effective.
>
> As shown below, merging a MergeLock-protected model results in **0% accuracy**, and *even with alignment* the performance remains **0%**, whereas PaRaMS is partially recoverable. This indicates that MergeLock generalizes robustly to large LLMs and variant attention designs.
>
> ### **LLaMA-8B-Instruct (math & code)**
>
> | Method | gsm8k | humaneval | avg |
> | --- | --- | --- | --- |
> | Base | 0.8196 | 0.7073 | 0.7635 |
> | code FT | 0.5732 | 0.7134 | 0.6433 |
> | math FT | 0.8438 | 0.3476 | 0.5957 |
> | TA (0.3) | 0.8393 | 0.4024 | 0.6209 |
> | **PaRaMS-protected math + code** | **0.0** | **0.0** | **0.0** |
> | **PaRaMS + alignment** | 0.7172 | 0.2378 | 0.4775 |
> | **MergeLock-protected math + code** | **0.0** | **0.0** | **0.0** |
> | **MergeLock + alignment** | **0.0** | **0.0** | **0.0** |
>
> While PaRaMS can be partially restored after alignment, MergeLock remains unrecoverable, confirming its stronger and more stable unmergeability on large and structurally diverse LLMs.
>
> **Q4: How easy would it be to estimate A and B matrices if the base and other fine-tuning model weights are known? i.e. there is the statement "such protection can be reversed by simply applying the inverse permutation"...**
>
> We appreciate the reviewer’s question. Unlike PaRaMS, whose protection relies mainly on permutation matrices (whose inverse is trivial), MergeLock uses general **invertible R·P·D transformations** that are neither purely discrete nor orthogonal, making estimation substantially more challenging.
>
> To assess recoverability, we attempted to invert MergeLock using the strongest available alignment technique—*“Beyond the Permutation Symmetry of Transformers: The Role of Rotation for Model Fusion” (ICML 2025)*—which reliably recovers permutation- and rotation-based transformations. However, under this method, the recovered mappings fail to approximate A/B, and the merged models remain unusable (Table 1 + LLaMA results).
>
> While we do not claim impossibility, **current alignment and symmetry-exploitation methods cannot effectively recover the MergeLock transformations**, suggesting that A/B are significantly harder to estimate compared to the permutation-only case in PaRaMS.
>
> **Q5: How expensive is this process of applying A and B?**
>
> The cost of constructing A and B is very small. Although each transformation is factored into R·P·D, all three components are **randomly sampled once**, without any optimization or iterative search. In practice, generating these matrices takes negligible time compared to fine-tuning or inference, and applying them is a single pass over the attention weights.
>
> Thus, MergeLock introduces **minimal computational overhead** while providing strong unmergeability.

---

> > ### Comment · Reviewer_jU9n · 2025-11-26
> > **Response**
> >
> > Thank you.
> >
> > These answer some of my concerns. I will increase the score.

---

> > > ### Author Response · Authors · 2025-11-27
> > >
> > > Thank you very much for your positive feedback and for increasing the score.
> > > If you have any further questions or would like clarification on any part of the paper, please feel free to let us know — we would be happy to provide additional details.

---

### Author Response · Authors · 2025-11-29
**Summary for AC after score rollback**

We would like to briefly summarize the key clarifications added during rebuttal and discussion.

First, several concerns regarding novelty and relation to PaRaMs were fully addressed: MergeLock exploits a distinct continuous symmetry family available only in self-attention (general invertible R-P-D transformations), does not require MLP modifications, and empirically yields substantially stronger unmergeability than permutation-based methods (1–5% vs. 70–95% recovery for PaRaMs).

Second, we added the large-model evaluations requested by the reviewers. On LLaMA-3/8B-Instruct (orders-of-magnitude larger and architecturally different from CLIP/Flan), MergeLock-protected models remain at 0% accuracy even after strong alignment, whereas PaRaMs partially recover (up to ~48%). This indicates MergeLock generalizes robustly across architectures, basin structures, and parameter scales.

Third, regarding recoverability and attack strength, we provided additional analysis showing that existing continuous-symmetry alignment methods (e.g., rotation-based recovery from ICML’25) fail to approximate the R-P-D mappings. We also clarified why stronger gradient-based or distillation-based attacks are unrealistic in the current open-source ecosystem.

Finally, we note that Reviewer jU9n explicitly stated after the rebuttal that the responses addressed their concerns and they raised their score. Due to the rollback, the publicly displayed score does not reflect this updated assessment. For Reviewer mTca, additional clarifications to Weaknesses 2 and 3 were posted after their last message, addressing practicality, key-sharing, and partial-alignment cases in detail.

We thank the AC for considering these clarifications and would be happy to provide any additional information.

---

### Meta-Review · Area_Chair_q8rd · 2026-01-06

**Summary:**

All three reviewers mentioned the limited/incremental novelty of MergeLock compared to prior work, particularly PaRaMS [jU9n, mTca]. Reviewers also commonly questioned the method's practicality/generalizability, noting that the current evaluation is insufficient, e.g., missing experiments on large decoder-only LLMs [K4CP, mTca, jU9n]. Furthermore, the defense against alignment seems to be a narrow threat model [mTca]. Other weaknesses include the large storage [mTca], and no significant new insights [jU9n].

**Reviewer Concerns:**

Some of the concerns were partially addressed by the authors with the newly provided results and clarification. However, concerns such as a large decoder-only LLM, it is unclear if the reviewers would deem the newly added result as large enough. Furthermore, Reviewer mTca’s concerns were not fully addressed. Specifically, the reviewer remains unconvinced by the practicality and evaluation.

**Reviewer Scores:**

Most likely, the positive reviewers would maintain their score. Reviewer mTCa would maintain their negative score, as the rebuttal did not fully address their concern. In the AC’s opinion, even after a full discussion period, it seems unlikely that the authors would be able to convince the reviewer.

---

### Decision · Program_Chairs · 2026-01-26

Reject